# Persistent Health and Cognitive Impairments up to Four Years Post-COVID-19 in Young Students: The Impact of Virus Variants and Vaccination Timing

**DOI:** 10.3390/biomedicines13010069

**Published:** 2024-12-30

**Authors:** Ashkan Latifi, Jaroslav Flegr

**Affiliations:** Laboratory of Evolutionary Biology, Department of Philosophy and History of Sciences, Faculty of Science, Charles University, Viničná 7, 128 00 Prague, Czech Republic; ashkan.latify@gmail.com

**Keywords:** SARS-CoV-2, cognition, mental health, long-term effects, long COVID

## Abstract

**Background:** The long-term consequences of COVID-19 infection are becoming increasingly evident in recent studies. This repeated cross-sectional study aimed to explore the long-term health and cognitive effects of COVID-19, focusing on how virus variants, vaccination, illness severity, and time since infection impact post-COVID-19 outcomes. **Methods:** We examined three cohorts of university students (*N* = 584) and used non-parametric methods to assess correlations of various health and cognitive variables with SARS-CoV-2 infection, COVID-19 severity, vaccination status, time since infection, time since vaccination, and virus variants. **Results:** Our results suggest that some health and cognitive impairments may persist, with some even appearing to progressively worsen—particularly fatigue in women and memory in men—up to four years post-infection. The data further indicate that the ancestral SARS-CoV-2 variant may have the most significant long-term impact, while the Omicron variant appears to have the least. Interestingly, the severity of the acute illness was not correlated with the variant of SARS-CoV-2. The analysis also revealed that individuals who contracted COVID-19 after vaccination had better health and cognitive outcomes compared to those infected before vaccination. **Conclusions**: Overall, our results indicate that even in young individuals who predominantly experienced only mild forms of the infection, a gradual decline in health and fitness can occur over a span of four years post-infection. Notably, some negative trends—at least in men—only began to stabilize or even reverse during the fourth year, whereas in women, these trends showed no such improvement. These findings suggest that the long-term public health impacts of COVID-19 may be more severe and affect a much broader population than is commonly assumed.

## 1. Introduction

The start of COVID-19, associated with a new coronavirus (SARS-CoV-2) found in Wuhan, China, occurred in late 2019, resulting in a worldwide pandemic [1,2]. While the acute effects of the virus were extensively studied in earlier phases of the pandemic, increasing evidence now highlights the potential for long-term impacts even after recovery. These impacts, commonly known as ‘long COVID’ or post-acute sequelae of SARS-CoV-2 infection, can manifest as a range of physical, neurological, and psychological symptoms [3,4]. Research has indicated that certain individuals may suffer a decrease in lung function or develop pulmonary fibrosis following recovery from COVID-19 [5]. In this respect, a study investigating the post-recovery results of 587,330 patients admitted to hospitals in the United States, of whom 257,075 had COVID-19 and 330,255 were not infected with the virus, found that among the patients, there were 10,979 cases of heart failure within a 367-day period after discharge. COVID-19 hospitalization led to a 45% increased risk of developing heart failure, particularly in younger patients, white individuals, and those with a prior history of heart conditions [6]. Similarly, a recent study of healthcare workers in Italy identified the severity of acute COVID-19, early pandemic waves (ancestral SARS-CoV-2 and Alpha variants), and prolonged viral shedding time as significant risk factors for long COVID-19. This study also highlighted that the prevalence of long COVID-19 decreased with subsequent infections and was lower during the Omicron wave, likely due to increased immunity and milder strains [7].

In a community-based cross-sectional study in China, researchers presented results for 1000 individuals who survived COVID-19 20 months after being diagnosed. These COVID-19 survivors were recruited from the communities in Hongshan district, Wuhan City, Hubei Province, from 12 October to 19 November 2021. The most commonly reported symptoms were aspecific (fatigue, joint pain, hair loss, sweating, myalgia, skin rash, and chill) (60.7%), mental (48.3%), cardio-pulmonary (39.8%), neurological (37.1%, including cognitive impairment in 15.6% of cases), and digestive (19.1%) symptoms [8]. A meta-analysis of 29 peer-reviewed studies and 4 preprints, published up until 20 March 2021, encompassing a sample of 15,244 hospitalized and 9011 non-hospitalized patients, revealed that 63.2%, 71.9%, and 45.9% of the sample exhibited at least one post-COVID-19 symptom 30, 60, or 90 days after the onset of illness or hospitalization. The most prevalent symptoms were fatigue and dyspnea. Additional reported post-COVID-19 symptoms included cough (20–25%), anosmia (10–20%), ageusia (15–20%), and joint pain (15–20%). Time trend analysis showed a decrease in symptom prevalence measured on day 30, followed by an increase measured on day 60 [9]. In this respect, a recent study, which conducted an online survey in South Korea between January and May 2022, surveyed 111 COVID-19 and 189 non-COVID-19 patients aged over 55, using tools like PRMQ, SMCQ, ISI (ISI-K), and WHOQOL-BREF (YLP-BREF). COVID-19 significantly impacted memory, physical activity, diet, depression, insomnia, and quality of life, but not anxiety, COVID-19 fear, or activity participation, highlighting critical differences between the groups [10].

Post-COVID-19 symptoms also encompass cognitive impairment and mental health deterioration. For instance, a large online survey of 4445 participants investigating the impact of SARS-CoV-2 infection, COVID-19 severity, and vaccination on health and cognition found that both infection with and the severity of COVID-19 had negative effects on patients’ health. Additionally, a correlation was observed between COVID-19 severity and various cognitive functions. Specifically, the severity of COVID-19 negatively impacted health and fatigue at the time of the study and had a significant adverse effect on intelligence and negative trends with memory, precision, and performance on the Stroop test [11]. A large-scale study of 81,337 participants in Britain also reported that individuals recovered from COVID-19, regardless of symptom presence, showed significant cognitive deficits in domains such as reasoning, problem solving, spatial planning, and target detection compared to controls after adjusting for various demographic and health factors. Hospitalized patients (*N* = 192) exhibited substantial deficits, as did non-hospitalized individuals with confirmed infections (*N* = 326). Analysis indicated that these cognitive differences were not pre-existing [12]. Another study, recruiting 475 participants (mean age 58.26) who were followed 2–3 years post-hospitalization, indicated that cognitive scores were significantly lower than expected across all domains assessed by the Montreal Cognitive Assessment—memory, executive functioning, attention, language, visuospatial skills, and orientation—with most participants reporting mild to severe depression, anxiety, fatigue, or cognitive decline. Symptoms worsened over time, and severity at 6 months strongly predicted outcomes at 2–3 years. Occupational changes were reported by 26.9% of participants, largely due to health issues, and were closely linked to cognitive deficits [13]. A recent systematic review of 16 studies (from 502 retrieved) examining COVID-19’s impact on working memory in patients without prior cognitive impairment also showed 22.5–55% experienced acute-phase impairment, with 6.2–10% still affected at six months. Factors influencing impairment included age, time since infection, and severity. Neuroinflammation, viral invasiveness, and brain structural changes were implicated, and concomitant symptoms often persisted long-term [14]

The effects of COVID-19 on young individuals after recovery can also include physical, cognitive, and psychological symptoms. Research indicates that some young subjects experience lingering symptoms which can include fatigue, cognitive difficulties, and respiratory issues even months after recovery [15]. A study of individuals from the Swedish birth cohort BAMSE (average age: 26.5 years) investigated post-COVID-19 symptoms, defined as those persisting for at least 2 months following confirmed infection. Altered smell and taste were the most frequent post-COVID-19 symptoms, followed by dyspnea and fatigue [16]. Another recent study assessing COVID-19’s effects on the physical and mental health, fatigue, and cognitive skills of a sample of 214 students averaging 21.8 years old showed that contracting COVID-19 was associated with a higher frequency of fatigue reports [17]. In addition, the disease’s impact on physical health, mental well-being, fatigue, and reaction time was moderated by severity. In the first 24 months following the infection, improvements in both physical and mental well-being were observed, along with a reduction in the frequency of errors in attention-demanding tests. However, there was a tendency towards a decline in reaction time and fatigue. By the second year, physical health and error rates improved, but fatigue and reaction times continued their ongoing decline, and mental health began to deteriorate.

More importantly, a meta-analysis of 55 studies involving 1,139,299 participants provided a broader view of post-COVID-19 conditions. This study found more than two hundred symptoms linked to this condition, with gastrointestinal issues, headaches, cough, and fever being the most common, respectively. From these studies, 21 symptoms reported in 11 studies were identified as suitable for inclusion in the meta-analysis, and their subsequent analysis revealed significantly higher pooled estimates for symptoms such as altered or lost smell or taste, dyspnea, fatigue, and myalgia in children and young people who had confirmed SARS-CoV-2 infections. The meta-analysis concluded that many children and young people experience persistent symptoms after SARS-CoV-2 infection [18]. Nonetheless, it should be noted that some studies suggest that ongoing symptoms and impairments in post-COVID-19 conditions may be influenced by factors beyond the virus itself, particularly psychosocial aspects of pre-existing psychological conditions [19]. For example, a study highlighted that pre-existing psychological conditions, such as depression and anxiety, are significant risk factors for persistent symptoms, including psychiatric and neurological complaints, as well as reduced work ability long after COVID-19 recovery [20].

It should be noted that many earlier studies have focused on a limited timeframe shortly after recovery from COVID-19 or overlooked the specific variant involved (see [21,22,23]). Extending the duration of observation, however, allows researchers to better track the potential long-term effects of post-COVID-19 infection on health and performance. The current article aimed to address these gaps along with the other objectives listed below.

### Aims and Scope of This Study

The primary aim of this study was to determine whether the trends observed in a previous, smaller-scale study (which included a sample size three times smaller and covered a shorter duration post-COVID-19 infection, specifically up to 39 months) continued into the fourth year after the initial illness (covering up to 54 months post-infection). The second objective was to rule out the possibility that the observed correlations between certain health- and performance-related variables and the time since infection were merely artifacts of the different virus variants, which emerged in distinct waves before the start of this study. The third aim of this study was to obtain a sufficiently large sample size to allow for the analysis of not only the impact of COVID-19 but also the effects of vaccination against COVID-19 and the duration since vaccination on health- and performance-related variables. A large sample size was essential to ensure adequate representation of the unvaccinated minority, comprising only about 5% of the student population, allowing for meaningful comparisons between vaccinated and unvaccinated groups.

To achieve these objectives in our cross-sectional study, we conducted tests on nearly 300 participants enrolled in an undergraduate evolutionary biology course each year for three consecutive years, consistently during the same month (January). The same panel of performance tests was administered, and data on current health status, details about their COVID-19 illness, and vaccination history were collected through a consistent electronic questionnaire. The collected data were analyzed using multivariate non-parametric methods, a choice informed by the statistical characteristics of the data. The findings, including those that may seem counterintuitive, are discussed considering the specifics of cross-sectional studies.

## 2. Materials and Methods

### 2.1. Study Participants

Over three consecutive years (2022, 2023, and 2024), always in the second week of January, all students enrolled in the basic course in evolutionary biology were invited to take part in an anonymous online study. However, it should be noted that the questionnaires were designed during the summer of 2021, and the protocol was approved by the IRB in the fall of 2021. This timing relatively limited the scope of the questions in accordance with the knowledge of COVID-19’s probable long-term effects available at that time. Before beginning the survey, participants were informed about the general goals of this study—examining specific hypotheses in evolutionary psychology and investigating the influence of various factors on cognitive performance. We included all students who provided their age and sex at birth, answered questions related to COVID-19 and health, and completed at least one of the cognition tests. The questionnaire distribution method excluded individuals who did not provide informed consent to participate in this study by clicking the appropriate button on the electronic form, as well as students who did not have proficiency in the Czech language.

The students were assured that their participation was voluntary and that the data would be used solely for scientific purposes, with no means of identifying individual participants. They were also informed that they could withdraw from this study at any time simply by closing the survey page.

Both the examination and the subsequent survey were conducted separately on the Qualtrics platform. After completing the exam, students were notified of their scores, specifically the number of correct answers, and were asked to report this number in the anonymous survey.

This study adhered to the ethical guidelines relevant to research involving human participants. The project’s protocol, participant recruitment methods, including the scope and generality of information provided to participants before this study, and the method of obtaining informed consent (achieved by clicking a designated button on the screen), received approval from the Institutional Review Board of the Faculty of Science at Charles University (No. 2021/19).

### 2.2. Questionnaires and Tests

In this survey, we assessed participants’ intelligence using the Cattell 16PF test (Variant A, Scale B) [24]. Memory abilities, including free recall and recognition memory, were evaluated using a modified Meili test [25,26]. Psychomotor skills, particularly reaction time and precision, were measured using the Choice Reaction Time Test and the Stroop Test, with both reaction time and accuracy recorded. Detailed descriptions of these tests are available elsewhere [17]. The ‘Reading Time’ variable was calculated as the average Z-score of the times taken to read instructions for the included tests. For each individual, the ‘Accuracy Score’ was calculated as the mean of the Z-scores of the number of correct answers in the four tests: Evolutionary Biology, IQ, Choice Reaction Time, and Stroop test. The ’Reaction Time Score’ was determined as the mean Z-score of Reading Time and reaction times recorded during the Choice Reaction Time test and Stroop test, while the ‘Memory Score’ was calculated as the mean Z-score of the two memory tests.

In the anamnestic section of the questionnaire, participants answered 21 questions regarding their physical health. These questions addressed the frequency of conditions such as allergies, skin disorders, cardiovascular diseases, digestive tract disorders, metabolic disorders, common infectious diseases, orthopedic disorders, neurological disorders, headaches, physical pains, and other chronic physical issues. Participants also provided information on how often they had taken antibiotics in the previous three years, their frequency of visits to a general practitioner, and the number of times they had been hospitalized for more than a week in the past five years. Responses to these questions were recorded on a 6-point Likert scale ranging from ‘never’ to ‘daily or more frequently’. Additionally, they reported the number of non-mental health medications currently prescribed by a doctor, with options ranging from 0 to 7, where 7 indicated six or more medications. At the start and end of the questionnaire, participants rated their current and usual physical well-being using 6-point scales. Finally, participants were asked to estimate their life expectancy, with six response options ranging from ‘more than 99 years’ to ‘less than 60 years’. They also provided information on their usual blood pressure, selecting from five options (‘very low’, ‘rather low’, ‘normal”, ‘rather high’, and ‘very high’). No participant selected ‘very high’, so we used these responses to calculate two binary variables: low blood pressure and high blood pressure. For the exact wording of all questions, refer to the questionnaire text attached to the pre-registration form (https://doi.org/10.17605/OSF.IO/M5FYC, accessed on 27 December 2024).

For health-related questions, certain responses were reversed so that higher scores consistently indicated poorer health, allowing for a standardized interpretation of the results. Then, a physical sickness index score was calculated using the mean Z-scores of all relevant questions [27]. Similarly, a mental sickness score was derived from questions related to nine variables, including frequencies of depression, anxiety, other mental health issues, the number of prescribed mental health medications, and assessments of the participant’s current and usual mental states. The latter two questions were posed at both the beginning and end of the questionnaire.

Participants were also asked to rate the frequency of fatigue using a 6-point scale ranging from ‘never’ to ‘daily or several times a day’, resulting in the variable Fatigue. Additionally, the survey collected demographic and medical history data, including participants’ age and official sex as listed on their birth certificates (with men coded as 1 and women as 0). At the end of the questionnaire, participants were queried about their history of SARS-CoV-2 infection. Responses to COVID-19 status were categorized as follows: ‘not yet’, ‘no but I was in quarantine’, ‘yes, I was diagnosed with COVID-19’, ‘probably yes, but I was not diagnosed with COVID-19’, and ‘I am waiting for the result of a diagnostic test’. Those who confirmed a COVID-19 diagnosis were asked to rate the severity of their illness on a 6-point scale, with 1 being ‘no symptoms’ and 6 being ‘I was in ICU’.

All participants provided information about their vaccination status, the date of their most severe COVID-19 episode, and the date of their last vaccination. These dates were used to calculate the time since the onset of COVID-19, the time since the last vaccination, and the binary variable COVID-19 after vaccination. Based on the dates of the waves of COVID-19 outbreaks in Czechia (see [28]) and the participants’ date of infection, the likely variant of SARS-CoV-2 was estimated for most participants and encoded into four binary dummy variables: ancestral SARS-CoV-2 variant, Alpha, Delta, and Omicron (for more information on the codes see Appendix A).

### 2.3. Data Analysis

To mitigate potential complications arising from the irregular distribution of some dependent variables and partly to avoid some of the issues associated with an imbalanced dataset—with a two-to-one ratio of women to men and vaccinated individuals outnumbering unvaccinated ones by more than tenfold—we employed a non-parametric multivariate analytical method, specifically the partial Kendall correlation controlled for age, sex, and survey year, to examine the relationships between variables related to COVID-19 (such as infection status, severity of COVID-19 infection, time elapsed since infection, vaccination status, time since the last vaccination, and occurrences of COVID-19 post-vaccination) and health and cognitive performance indicators. In analyses specific to sex, only age and survey year were controlled. The partial Kendall correlation offers several advantages that make it particularly suitable for this study. Being an exact test, it enables precise significance calculations even in cases where some groups contain very few observations, such as unvaccinated men. Additionally, its ability to analyze associations between binary, ordinal, and continuous variables simplifies the analytical process. This avoids the need for multiple distinct methods, thereby enhancing the clarity and consistency of the results presentation.

Furthermore, the partial Kendall correlation can control for the effects of virtually any number of covariates of various types (nominal categorical via dummy coding, ordinal as ranks, and continuous variables directly) if a sufficiently powerful computational setup is available. While the test is theoretically less sensitive than parametric methods due to its reliance on variable ranks rather than raw values, it performs comparably well with real-world biological data, which seldom adhere to a Gaussian distribution. This combination of precision, versatility, and robust performance with non-normally distributed data makes partial Kendall correlation an optimal choice for addressing the analytical challenges posed by our dataset.

The correlational analysis was conducted using the Explorer v. 1.0 R script [29], leveraging the ppcor R package v. 1.1 [30].

To correct for multiple comparisons, we applied the Benjamini–Hochberg correction for multiple testing with a false discovery rate (FDR) set at 0.10 [31]. This correction was performed only for the main seven variables (physical health issues, mental health issues, fatigue, intelligence, memory, reactions, accuracy), not for post hoc tests exploring the sources of observed associations. In addition, due to the issue of normality, we employed the Wilcoxon rank-sum test to compare the frequency, age, and the time elapsed since infection between men and women in the dataset. The dataset for this study is publicly accessible on Figshare 10.6084/m9.figshare.27618876 [32].

Technical Notes: In this article, the term ‘effect’ is used in its statistical sense, referring to the difference between the true population parameter and the null hypothesis value. Causal interpretations are only discussed in the Discussion section. Given the exploratory nature of a large part of this study, we also consider trends that did not reach statistical significance.

## 3. Results

### 3.1. Descriptive Statistics

The initial dataset comprised 724 individuals, 520 women (71.8%, average age = 21.54, SD = 2.36) and 204 men (28.2%; average age = 21.50; SD = 2.13). Among the women, 169 (36.18%) reported not having been diagnosed with COVID-19, 210 (45.0%) had a confirmed laboratory diagnosis of COVID-19, 52 (11.1%) believed they had contracted the virus but lacked laboratory confirmation, and 36 (7.7%) reported not having COVID-19 but were quarantined. Among the men, 61 (32.4%) reported not having been diagnosed with COVID-19, 85 (45.2%) had a confirmed laboratory diagnosis, 19 (10.1%) believed they had contracted the virus but lacked laboratory confirmation, and 23 (12.2%) reported not having COVID-19 but were quarantined. No students reported awaiting test results.

From the final dataset, we excluded 52 individuals who stated they might have had COVID-19 but lacked laboratory confirmation, 156 individuals who did not respond to COVID-19-related questions, and 4 individuals who provided unreliable data, such as reporting an age over 90 or giving identical responses to most questions. The final sample included 584 participants: 415 women (49.4% COVID-19-negative, 50.6% COVID-19-positive) and 169 men (49.7% COVID-19-negative, 50.3% COVID-19-positive). There was no significant difference in the incidence of COVID-19 between men and women (Chi^2^_(1)_ = 0.004; *p* = 0.946). The ages of the women and men were similar (21.58, SD 2.36 vs. 21.51, SD 2.02; *W* = 33945; *p* = 0.624). Similarly, there was no significant age difference between uninfected and infected individuals (21.59, SD 1.99 vs. 21.53 SD 2.51; *W* = 44699; *p* = 0.200).

A significant majority of the students were vaccinated against COVID-19, with the last vaccination occurring on average 12.38 months prior and at most 50.8 months prior. Among 409 women who responded to this question, 34 (8.3%) were not vaccinated; among 165 men, only 7 (4.2%) were unvaccinated; however, this difference was not statistically significant (Chi^2^_(1)_ = 2.93; *p* = 0.086). Of the 100 vaccinated women who had COVID-19, 29 (29.0%) contracted the virus after their last vaccination; among 49 men, this was the case for 17 (34.7%), (Chi^2^_(1)_ = 0.499; *p* = 0.479).

On average, women contracted COVID-19 24.39 months prior (SD = 14.32; maximum 54.58), and men 21.68 months prior (SD 13.97; maximum = 52.61). In this respect, there was no significant difference between these two groups (*W* = 0.5368; *p* = 0.261). Among the 282 individuals diagnosed with COVID-19, 26 (9.22%) reported ‘No symptoms’, 94 (33.33%) described it as ‘Like mild flu’, another 106 (37.58%) as ‘Like normal flu’, and 52 (18.44%) as ‘Like severe flu’; 4 (1.41%) were hospitalized. None reported being treated in the Intensive Care Unit (ICU). The distribution of responses regarding the severity of COVID-19 did not differ between men and women (Chi^2^_(4)_ = 1.47; *p* = 0.831). In contrast, the long-term impact of COVID-19 on health and performance, particularly the dynamics of these changes over time, differed by sex (see below). Accordingly, all relevant analyses were stratified by sex, and descriptive statistics (Table 1 and Appendix A) were presented separately for men and women.

### 3.2. Confirmation Statistics—Associations of COVID-19-Related Variables with Health and Cognition Issues

In the confirmatory part of the study, we aimed to test the following H_0_ hypotheses:

**H1:** 
*Trends in health and cognitive performance impairments do not persist into the fourth year after infection.*


**H2:** 
*The observed trends over time are artifacts caused by the gradual replacement of more virulent strains with less and less virulent variants of SARS-CoV-2.*


**H3:** 
*There is no difference in health and cognitive performance between individuals vaccinated before or after contracting COVID-19.*


**H4:** 
*Neither the health nor cognitive performance of participants changes with the time elapsed since vaccination.*


Age and especially sex significantly impacted some health- and performance-related variables. Consequently, we used a partial Kendall correlation, controlling for age, sex, and the survey year, to investigate the effects of COVID-19-related variables on health and performance outcomes (Table 2). The partial correlations between health and performance outcomes associated with different variants of SARS-CoV-2 are presented separately (Table 3).

Our analysis showed that the severity of COVID-19 was significantly associated with worse physical health, fatigue, and a lower cognitive accuracy. After applying the Benjamini–Hochberg (BH) correction for multiple testing, all of these associations remained significant. COVID-19 vaccination was associated with poorer physical health and increased fatigue but with improved (shorter) reaction times in vaccinated students compared to unvaccinated ones. All of these associations remained significant even after the BH correction. Subjects who contracted COVID-19 after receiving the vaccine showed significantly higher intelligence and accuracy scores than those who contracted COVID-19 before receiving the vaccine. These also remained significant after the BH correction. The analysis of the effects of virus variants revealed that the ancestral SARS-CoV-2 variant had the most adverse effects compared to other variants. This variant correlated with lower scores in the subjects’ intelligence and memory and higher scores in their reaction times, which mean worse cognitive performance, and was associated with higher levels of fatigue. These associations remained significant even after applying the BH correction. The Alpha variant demonstrated significant negative associations with fatigue and reaction times, indicating lower levels of fatigue and also shorter (better) reaction times compared to other variants. The Omicron variant exhibited a significant positive association with the subjects’ intelligence scores, indicating higher intelligence scores in those infected with this variant. The Delta variant did not have any significantly negative or positive associations with any of these seven variables.

There were no significant associations between the seven index variables and COVID-19 infection, the time elapsed since COVID-19 infection, or the time elapsed since receiving COVID-19 vaccination (see Table 2). The results of separate analyses for women and men are shown in Appendix A.

When we included the variable representing the SARS-CoV-2 variant in the model (coded ordinally, with 1 for the earliest variant, the ancestral SARS-CoV-2 variant, and 4 for the most recent variant, Omicron), the results remained unchanged (see Appendix A). This result suggests that the observed patterns of effects and trends and the absence of trends among the predictors were not simply artifacts caused by the replacement of more virulent variants with progressively less virulent ones throughout the course of the pandemic.

Our analyses of the trajectories of the index variables over the time elapsed since COVID-19 infection (almost 54 months) for all participants, and also separately for men and women, indicated that the first-degree linear models had smaller Bayesian Information Criterion (BIC) coefficients compared to those of higher-degree polynomial models. Accordingly, the following section discusses the results based on our first-degree linear models.

The results for all participants showed a noticeable, relatively small decline in memory performance over the time elapsed since infection. In contrast, trends of marginal improvement in mental health issues, accuracy, and intelligence were observed. Fatigue showed the largest increase over the time elapsed since infection. Marginal trends of deterioration in reaction times and physical health were also present (see Figure 1 and Table 2). The results of separate analyses for women and men are shown in Appendix A.

The results of the previous study [17] suggested that the direction and rate of changes over time varied during the first two years after infection but likely increased thereafter. Therefore, we repeated the analyses for 111 subjects from 24 months post-infection onwards. Figure 2 and Appendix A show that there were negative associations between months since infection and physical health, mental health, fatigue, intelligence, memory, reaction times, and accuracy during the fourth year after infection in this group, among which the correlation between intelligence and months since infection was significant. Based on the binomial distribution, the probability of obtaining seven correlations indicating adverse relationships out of seven would be 0.008. See Appendix A for separate analyses of the trajectories of the seven index variables over time beyond 24 months post-infection for women and men, respectively. When fifty-one participants who were infected at least 36 months before this study were analyzed, the results were more complex (Appendix A). In the whole population, we saw a relative decrease in intelligence (Tau = −0.072), and worsening of reaction times (Tau = 0.093); however, we also saw a clear improvement in accuracy (Tau = 0.125). Analyses split by sex showed that in women, all health- and cognitive performance-related variables except accuracy (Tau = 0.144) and memory (Tau = 0.038) were impaired (physical issues: Tau = 0.077; mental issues: Tau = 0.213; fatigue: Tau = 0.083; intelligence: Tau = −0.079; and reactions: Tau = 0.054). In contrast, in men, all changes except worsening in reaction times (Tau = 0.350) were improvements (physical issues: Tau = −0.328; mental issues: Tau = −0.414; fatigue: Tau = −0.159; intelligence: Tau = 0.129; memory: Tau = 0.314; and accuracy: Tau = 0.060). It must be noted, however, that none of the trends observed in participants infected more than three years ago were statistically significant, and only 15 participants were included in the subset of men (for the correlational analyses split by sex, see Appendix A for women and Appendix A for men, respectively).

When we analyzed the effect of time since vaccination, as opposed to time since COVID-19 infection, we observed rather distinct results (Figure 3). The comparison of results in columns 3 and 5 of Table 2 shows that correlations with time since vaccination were consistently weaker than those with time since COVID-19. In the case of accuracy and intelligence, the correlations even shifted from negative (indicating worsening performance over time) to positive. None of the correlations came anywhere close to the threshold for statistical significance (that is, *p* < 0.05). See Appendix A for separate analyses of the trajectories of the seven index variables over the months following vaccination for women and men, respectively.

A summary of the results of the confirmatory part of this study is provided below.

We rejected hypotheses H1 and H3, but we could not reject hypotheses H2 and H4. In other words, our data suggest the following:(1)The deterioration of certain health and cognitive functions continues even during the fourth year after COVID-19. However, the direction of nonsignificant trends in the subset of 15 men more than three years post-COVID-19 suggested that the deterioration may stop and potentially even reverse, except for intelligence and reaction times, at least in men.(2)To test hypothesis H2, we also included the confounding factor ‘COVID-19 variant’ in the model. The results showed that the lack of significant correlations between the outcome variables and the time since infection persisted, even with this factor in the model. However, all nonsignificant trends were markedly weaker when the confounding variable of the virus variant was controlled. In several cases, the corresponding trend even reversed direction. For example, the Tau value characterizing memory decline shifted from −0.053 to 0.006, and in the case of intelligence, from −0.071 to 0.012 (see Appendix A and compare with Appendix A). Thus, based on our data, we cannot refute hypothesis H2, which posits that the observed nonsignificant trends (the intensification of certain symptoms of impaired health or cognition over time since infection) are artifacts caused by the gradual replacement of more virulent strains with progressively less virulent variants of SARS-CoV-2.(3)Additionally, individuals who were vaccinated before contracting COVID-19 are in better health and have better cognitive performance than those who received the vaccination only after recovering from COVID-19.(4)Finally, neither the health nor the cognitive performance of participants changed with the time elapsed since vaccination.

### 3.3. Exploration Statistics—Association of COVID-19-Related Variables with Primary Health- and Cognition-Related Variables

In this exploratory part of this study, we did not apply corrections for multiple testing [33].

#### 3.3.1. All Participants

Generally, the severity of COVID-19 (course of COVID-19) showed strong, statistically significant positive associations with worse health outcomes and lower cognitive test performance; see Appendix A. These included lower precision in Stroop test performance, higher levels of allergies, skin conditions, digestive issues, infections, neurological problems, headaches, recurrent health issues, overall physical pain, depression, overall psychological problems, higher consumption of psychotropic medication, higher rates of doctor visits, higher use of other types of medication, greater antibiotic intake over the past three years, more frequent hospitalizations in the past five years, and shorter life expectancy.

In contrast, the associations of health and performance with merely having contracted COVID-19 were much less frequent and generally weaker and nonsignificant. Exceptions included a significant negative association with usually feeling mentally miserable, and positive associations with recognition memory, neurological disorders, infections, and the frequency of antibiotic use in the past three years. The last two variables included COVID-19 and the treatment of associated bacterial infections, making their correlation with contracting COVID-19 inevitable. Individuals who had contracted COVID-19 reported generally feeling mentally well.

COVID-19 vaccination demonstrated positive associations with allergies, digestive problems, cardiovascular issues, headaches, recurrent health problems, medication intake, and usually feeling mentally miserable. However, it was associated with shorter (better) simple reaction times.

Individuals who contracted COVID-19 after vaccination exhibited higher scores on the evolutionary biology exam and better Stroop test precision. They also had higher rates of skin conditions, elevated blood pressure, and reported a longer life expectancy compared to those infected before vaccination.

To assess the differences in the effects of various virus variants, we studied the correlations of the output variables with four dummy variables, each indicating whether a person was infected with a specific variant (coded as 1) compared with other variants (coded as 0). Investigating the correlations of the output variables with these dummy variables showed that infection with the ancestral SARS-CoV-2 variant was negatively associated with memory (it had negative associations with both free-recall memory and recognition memory). It was also linked to higher rates of recurrent health problems. Infection with the Alpha variant was associated with worse precision in simple reaction time tests, lower instances of infections, higher rates of orthopedic issues, and a higher tendency to feel mentally miserable. The Delta variant did not show significant associations with the output variables, while the Omicron variant was linked to higher levels of depression and anxiety.

Associations between the studied output variables and the time since infection were weak and sparse, and even weaker were corresponding associations between these variables and time since vaccination.

Results of all analyses remained nearly the same when we also controlled for the time elapsed since infection in addition to age, sex, and survey year (see Appendix A), showcasing that the associations of virus variants with health and cognition were not confounded by this variable.

#### 3.3.2. Women

Investigating the results for women and men separately revealed several statistically significant associations. A comprehensive set of results is presented in Appendix A.

In the female group, the long-term effects of COVID-19 infection were evident in higher rates of allergies, higher susceptibility to infections, a higher number of doctor visits, and an elevated rate of antibiotic consumption over the past three years. Conversely, they expressed lower psychological distress, as indicated by a reduced tendency to usually feel mentally miserable, along with higher memory and word recognition scores. More critically, the severity of COVID-19 was associated with a greater incidence of allergies, skin conditions, digestive issues, infections, neurological problems, and headaches. It also correlated with more frequent physical problems, a higher number of doctor visits, and higher rates of antibiotic use during the past three years, as well as greater medication consumption for both mental and physical health concerns. The analysis for this group also showed that there were negative associations between the time elapsed since infection and both the utilization of psychotropic medications and the number of doctor visits.

COVID-19 vaccination was associated with better performance in simple reaction times tests, higher rates of allergies, cardiovascular issues, orthopedic problems, headaches, medication consumption, a tendency to feel mentally miserable, and shorter life expectancy in the female group. Additionally, there was a positive correlation between the time elapsed since COVID-19 vaccination and higher precision on the Stroop test. Those who contracted COVID-19 after vaccination had higher scores on the evolutionary biology exam and better precision on the Stroop test. They also reported higher rates of skin conditions, infections, neurological problems, and feeling physically miserable, but reported a longer life expectancy.

We also investigated the associations between the four variants of the SARS-CoV-2 virus and the output variables. The analysis showed that infection with the ancestral SARS-CoV-2 variant was associated with more frequent orthopedic problems, overall physical pains, and recurrent health issues. Infection with the Alpha variant was associated with lower simple reaction time precision, lower precision on the Stroop test, fewer skin conditions, lower rates of infections, and reduced levels of medication consumption. Infection with the Delta variant was associated with higher rates of metabolic problems but lower levels of anxiety. And finally, infection with the Omicron variant was associated with lower scores on the evolutionary biology exam, better performance on the Stroop test, lower orthopedic issues, higher levels of depression, higher levels of anxiety, and higher rates of the utilization of medication for mental issue purposes.

#### 3.3.3. Men

Despite the small sample size of men, COVID-19 was significantly associated with worse reading time and more frequent hospitalization over the past five years but lower levels of medication use for mental purposes and overall less frequent other psychological problems. More importantly, the severity of COVID-19 was associated with higher rates of allergies, metabolic problems, recurrent health conditions, overall other physical pains, overall other psychological problems, higher utilization of medications for mental health purposes, a larger number of doctor visits, higher overall medication consumption, higher incidents of hospitalization during the past five years, and worse precision on the Stroop test. The time elapsed since infection did not show any significant correlations with the output variables, but the time since vaccination was associated with reduced levels of depression and lower mental distress at the time of the survey. Those who had contracted COVID-19 after vaccination in this group had lower levels of feeling physically miserable now than those who had contracted COVID-19 before vaccination. COVID-19 vaccination was associated with higher rates of infections, headaches, lower incidents of hospitalizations during the past five years, and lower blood pressure.

Investigating the effects of the four variants of the SARS-CoV-2 virus indicated that infection with the ancestral SARS-CoV-2 variant was associated with worse performance in both memory tests, lower orthopedic issues, and lower anxiety. Infection with Alpha variant was associated with worse performance in the recognition memory task, shorter reading times, higher cardiovascular problems, more orthopedic issues, more headaches, higher rates of doctor visits, and longer life expectancies. Infection with the Delta variant was associated with better performance in both memory tests, lower doctor visits, and lower consumption of medications. Infection with the Omicron variant was associated with better performance on the evolutionary biology exam, lower levels of feeling mentally miserable now, and a negative association with low blood pressure. The aforementioned results concerning the long-term consequences of COVID-19 infection differed from those related to the acute phase of the illness. Appendix A show that no effect of any SARS-CoV-2 variant on the reported severity of acute COVID-19 was detected.

## 4. Discussion

The current study employed a design and technical execution similar to our previous research [17]. However, it differed in that the data for this study were collected from participants of a basic-level course in evolutionary biology, which had twice the number of students compared to the advanced course used in the previous study. This larger student population allowed for a more robust sample size. Data collection occurred each time in January instead of May and spanned the years 2022, 2023, and 2024, whereas the previous study was limited to 2022 and 2023. Additionally, the questionnaire used in this study differed slightly from the previous one, with most changes affecting questions unrelated to the core objectives of either study. The current study aimed to determine whether the health and performance trends observed over 39 months post-infection in the previous study persisted mainly in the following 12 months. Furthermore, the current study sought to determine whether these trends were due to increasing time since infection or merely because the time since infection closely correlated with the virus variant a person was infected with. Lastly, the nearly threefold increase in respondents in the new study (584 vs. 214) allowed for the analysis of the impact of vaccination, which only a small number of students had not received (8.3% of women and 4.2% of men).

### 4.1. Trends in Health and Performance Post-COVID-19

The results showed that some trends reversed, with health and performance parameters beginning to improve, although some negative trends persisted even 51 months post-COVID-19. This was particularly evident in the higher levels of fatigue among both men and women. At least in men, there was a continued decline in physical health. While previous data from a smaller dataset suggested that the direction of ongoing changes varied over time, the new, more extensive data indicated that the trends were likely linear, as the relevant data points were best fitted by a first-degree polynomial, a straight line. Even so, beyond three years post-COVID-19 infection, mental health, memory, intelligence, fatigue, reaction times, and accuracy deteriorated further; however, physical health remained relatively constant. When men who had been infected more than 3 years prior were analyzed separately, it appeared that the negative trends were finally beginning to reverse. However, given the small number of these men (15), this result should be interpreted with caution, and it will certainly need to be confirmed in a larger sample.

In our study, we interpreted the observed trends of performance decline over time since infection as either a result of the gradual accumulation of the infection’s negative effects or, alternatively, as the replacement of more virulent strains by less virulent ones over the course of the epidemic. However, due to the design of this study, all participants were approximately 21.5 years old. As a result, the time since infection negatively correlated with the age at which each participant contracted COVID-19. It is therefore possible that the poorer health and cognitive performance in individuals who had COVID-19 longer ago could be related to the fact that they were younger at the time of infection, and their bodies, particularly their brains, were more susceptible to the negative effects of the illness. Recent studies in part corroborate this suggestion and warrant further research [34,35].

### 4.2. Impact of SARS-CoV-2 Variants

Our study confirmed that the SARS-CoV-2 variant affected the variables studied, similar to findings by [36,37]. Similar to the study by Cegelon et al. [7], our study also demonstrated that the ancestral SARS-CoV-2 variant had the most severe long-term consequences, while the Omicron variant had the mildest. However, these studies focused mainly on physical health variables or case fatality rates, whereas our research examined a broader range of health and cognitive outcomes. Specifically, Torabi et al. almost exclusively focused on physical health variables such as myalgia, cough, taste/smell disorder, diarrhea, etc., and investigated the personal characteristics, symptoms, and underlying conditions of individuals infected with different variants of SARS-CoV-2 over a period of two years, while Xia et al. concentrated on the case fatality rates of the variants of this virus during the epidemic periods by continents. In our study, the observed associations between the studied dependent variables and the dummy variables representing the different virus variants did not change when the time since infection was included as a covariate in Kendall’s partial correlation tests, save for the removal of the effects of the ancestral SARS-CoV-2 variant on the three variables related to intelligence, reactions, and memory. This indicates that the observed differences are primarily driven by the specific virus variant present at the time of infection, rather than the time elapsed since infection. Conversely, the noticeable changes in the model outputs after controlling for the ordinal virus variant variable suggest that the trends—namely, the weakening or intensifying of symptoms over time since infection—could be driven by differences in the virulence levels of the virus variants contracted at different times.

Our data do not clarify whether the more pronounced symptoms associated with the ancestral SARS-CoV-2 variant are due to its inherent properties or the fact that, during its spread, there was no vaccine available in the Czech Republic, leading to all patients infected with this variant being unvaccinated. However, the absence of vaccination cannot fully account for these differences, as most students infected with the Alpha variant also had not been vaccinated, given that the vaccination of young, low-risk individuals in the Czech Republic began only in the summer months of 2021, i.e., after the Alpha variant wave had passed. This suggests that factors beyond vaccination status likely contributed to the differences in symptoms between the ancestral SARS-CoV-2 variant and other variants. An unexpected result of this study is that the observed differences between the virus variants were limited to the long-term consequences of COVID-19 and had no measurable effect on the severity of the acute phase of the illness.

When discussing the varying effects of different virus variants, it is important to reiterate that our study was conducted on a young student population, where the illness generally presented mildly, as evidenced by the fact that only four students (0.7%) required hospitalization due to COVID-19. It is possible, and indeed likely, that the impact of different virus variants on individuals in older age groups, particularly those who experienced severe or very severe illness, would differ significantly from our observations.

### 4.3. Impact of Sex

We did not observe any differences in the risk of infection or the severity of COVID-19 between men and women. However, men and women differed significantly in physical and mental health, as well as in cognition-related variables. With the exception of memory and reaction times, women scored significantly worse on all seven focal variables (see Appendix A). This was the primary reason for controlling for sex in the confirmatory part of this study and for analyzing the effects of COVID-19-related variables on health and cognition separately for men and women in the main exploratory analysis. This approach, however, did not allow for formal testing of the effect of the interaction between sex and COVID-19-related variables on health and performance. Nevertheless, visual inspection of Appendix A suggests notable trends. Among individuals who had recovered from COVID-19, men reported significantly better physical and mental health immediately after infection. Over time, however, these measures showed substantial declines. In contrast, women’s physical health remained relatively stable throughout the observation period, while their mental health gradually improved.

Men reported lower fatigue levels than women shortly after infection, but fatigue in men progressively worsened over time. In women, fatigue levels also worsened, but at a faster rate, resulting in a pronounced difference between the sexes by the end of the study period. Intelligence in men was slightly higher shortly after infection but gradually declined, while intelligence in women showed a slight increase over time, eventually surpassing that of men by the end of the observation period.

Memory was markedly poorer in men immediately after infection and declined at a much faster rate, leading to a difference between men and women that was twice as large at the end of the study period compared to its beginning. In contrast, accuracy in cognitive performance tests was slightly higher in men immediately post-infection and increased markedly more over time than in women, leading to a notable advantage for men by the end of the study. Reaction times were initially slightly better in men and gradually worsened at a similar rate to those in women.

The most pronounced differences emerged during the fourth year post-infection. While women continued to experience declines in health and cognitive performance (except for memory and accuracy in cognitive tests), men showed improvement in all parameters except reaction times. However, it is important to note that the dataset included only 15 men who had been infected more than three years prior, which limits the generalizability of these observations.

### 4.4. Role of Vaccination in Health and Performance

As previously mentioned, individuals who had been vaccinated showed poorer health outcomes in some aspects compared to unvaccinated individuals. As suggested in earlier studies [11,17], the most probable explanation for this phenomenon is that individuals with poorer health were more likely to opt for vaccination as a precaution during the pandemic. This explanation is substantiated by a number of studies showing that the elderly and those at risk held more positive attitudes towards vaccination [38,39]. In our study, we further supported this hypothesis by finding no significant correlations between health- and performance-related variables and the time elapsed since vaccination. Even when nonsignificant trends were present, they were generally much weaker than those related to the time since COVID-19 infection. It is also important to note that, in contrast to the frequent adverse correlations with time since COVID-19, nearly all correlations with time since vaccination were beneficial.

Our data corroborate earlier findings that the experience of having had COVID-19 has a relatively weak impact on the health status and cognitive performance of young individuals [11,17,40]. On average, those who had been infected demonstrated slightly better health and performed better on performance tests. The most pronounced and often significant negative symptoms were observed in individuals previously infected with the ancestral SARS-CoV-2 variant, while those infected with the Omicron variant showed the least severe symptoms. However, even in our sample of young individuals, the severity of the illness had a strong impact. The intensity of symptoms associated with worsening health generally changed only slowly with time since COVID-19 illness (mostly showing nonsignificant trends). *Vaccinated individuals generally exhibited poorer health compared to unvaccinated individuals. However, the difference in their health and performance did not change significantly with time since vaccination, which does not support the existence of a causal relationship between vaccination and poorer health. Most importantly, individuals vaccinated before contracting COVID-19 demonstrated superior performance on cognitive tests and often exhibited significantly (or nonsignificantly) better health parameters compared to those vaccinated after recovering from COVID-19.* Of course, the difference in health between these two subgroups (individuals vaccinated before and after contracting COVID-19) could have existed prior to vaccination, as high-risk individuals were prioritized for vaccination during the early phases of the pandemic, unlike low-risk individuals. However, if this were the case, we would expect those in worse health to have been vaccinated before, not after, contracting COVID-19.

The relatively weak impacts of having had COVID-19, as well as the observed negative association between vaccination and health status, can be attributed to the design of this study. Since this was an observational cross-sectional study, individuals were not randomly assigned to groups such as infected vs. uninfected, vaccinated vs. unvaccinated, or vaccinated before vs. after contracting COVID-19. Previous studies have shown that individuals with better health were more likely to contract COVID-19, as they took fewer precautions against infection, compared to those with chronic health issues [41,42]. Indeed, this inverse relationship between long-term health conditions and the likelihood of contracting the infection was statistically confirmed [11,43]. A study that separately tracked long-term and current health statuses showed a nonsignificant negative association between prior health issues and contracting COVID-19 and a significant positive association between contracting COVID-19 and current health problems [11]. The same study revealed a nonsignificant negative association between vaccination and current health issues and a significant positive association with long-term health problems. Unlike the present study, the earlier study included 4445 individuals from a general internet population. This population was much more age-diverse, allowing for the study of age effects. As expected, age was much more strongly correlated with health and cognitive performance than in our highly age-homogeneous sample. The study showed that younger individuals were more likely to contract COVID-19 and less likely to be vaccinated compared to older individuals, who were generally more at risk. The positive association between long-term poor health and the willingness to get vaccinated persisted even after controlling for age, suggesting that the reported link between vaccination willingness and age is not mediated by age itself, but by age-related health issues.

For a precise examination of the impact of COVID-19 and vaccination on health status and cognitive performance, data from randomized experimental studies would be necessary. However, these studies are not feasible, except in rare cases, due to ethical constraints. A potential solution is to monitor the severity of the illness, the time elapsed since infection, and the time elapsed since vaccination, as we did in this and our previous study [17]. Additionally, prospective observational studies can provide important information regarding the causality of observed associations. However, prospective studies are more demanding to conduct, require large sample sizes, and involve repeated assessments of the same individuals before and after illness or vaccination. A prospective study conducted on 30,000 Czech internet users, of whom 5164 participated repeatedly, showed that having COVID-19 had a strong impact on physical health (partial Tau = 0.25) and a weaker yet still highly significant impact (partial Tau = 0.04, *p* < 0.001) on mental health [43]. Data collection for this study was completed by the end of March 2021, so the study could not assess the impact of vaccines on health status.

### 4.5. Strength and Limitations

The present study has several strengths and limitations. One of its main strengths, compared to similar studies, is the rigorous implementation of all available measures to limit selection bias and reduce distortions arising from subjective opinions on the topic. More than 80% of course participants volunteered for this study. Given the strong polarization within Czech society regarding the effects of COVID-19 and vaccination on health, steps were taken to avoid potential bias stemming from participants’ personal beliefs. For example, students were informed that this study would investigate the impact of various ’biological factors’ on health, but they were not explicitly told that COVID-19 infection would be one of these factors. Additionally, relevant questions about COVID-19 and vaccination were placed at the end of the questionnaire, ensuring that participants answered health-related questions and completed performance tests without being given any indication that this study might relate to COVID-19.

However, this study also had some limitations. The first is an inherent limitation of all cross-sectional observational studies: the observed statistical associations do not establish causality and cannot discern what is cause and what is effect. The basic design of our study was a serial cross-sectional study. This design allows for the identification of statistical associations between dependent and independent variables. However, without supplementary information from other studies, it does not allow for conclusions about causality.

For example, the observed better health status of individuals who had experienced COVID-19 could be explained in two ways: either that contracting COVID-19 had a positive effect on health or that individuals in better health were less cautious in protecting themselves from SARS-CoV-2 infection during the first year of the pandemic, prior to the availability of vaccines, and thus were more likely to contract COVID-19. Similarly, the observed associations between health, cognitive performance, and vaccination could be interpreted in dramatically different ways.

To assess which interpretation of the observed associations is more plausible, we applied the established Bradford Hill criteria for causality in epidemiology [39]. Specifically, we focused on the following:

Strength of Association (Criterion 1): The more severe a case of COVID-19 a student experienced, the poorer their health status and cognitive performance at the time of the study.

Temporality (Criterion 4): Individuals exhibited better health and cognitive performance if they were vaccinated before contracting COVID-19 compared to those vaccinated after contracting the virus.

Plausibility (Criterion 6): It is more likely that SARS-CoV-2 infection worsens health rather than improves it.

It is important to emphasize, however, that these criteria are merely heuristic tools. A definitive answer to questions of causality can only be provided by experimental studies—which, in the case of COVID-19, are not ethically feasible. Accordingly, we have maintained a moderate tone in the causal interpretation of our findings. Furthermore, we acknowledge that some of the purported health or performance effects associated with long COVID may not stem solely from COVID-19-induced complications but could also arise from other factors such as psychological issues or individual vulnerabilities to health problems that may or may not be related to COVID-19. Nevertheless, the design of our study and the associations identified at the sample level suggest that these effects are more likely attributable to long COVID rather than to individual vulnerabilities unrelated to the effects of COVID-19.

Another limitation of this study was that the assessment of participants’ health relied on their self-reported information. To minimize the subjectivity inherent in self-assessment, we required participants to answer 30 questions addressing specific aspects of their health (e.g., the number of different prescription medications they were currently taking, the number of hospitalizations lasting more than a week in the past five years, and similar metrics). Based on their responses to these questions, we calculated indices of physical and mental health. Cognitive performance, on the other hand, was directly measured as part of this study.

In this study, we controlled for only three variables: age, survey year, and sex. Naturally, both health and cognitive performance are influenced by numerous additional confounding factors that were not examined or included in our statistical models. It is important to emphasize that the population of the Czech Republic, and particularly the participants in this study, is exceptionally homogeneous. Nearly all participants resided in Prague, all shared a Caucasian European racial background, and all belonged to a similar age group. They also had comparable educational qualifications as undergraduate biology students enrolled in an elective course on evolutionary biology, likely driven by similar motivations and interests. This high degree of homogeneity significantly minimized variability that could introduce confounding effects. However, it simultaneously limits the generalizability of the findings.

A further limitation was technical: the questionnaire asked participants for the date of their last vaccination, which, along with the date of their most severe COVID-19 episode, was used to determine the binary variable ‘COVID-19 before or after vaccination.’ In reality, some participants with a longer time since their last COVID-19 infection than since their last vaccination had their first vaccination before contracting COVID-19. The presence of such misclassified individuals in the dataset likely skewed the results, specifically diminishing the measured effects of the binary variable vaccination before COVID-19 and increasing the risk of failing to detect an existing effect (but not risk of detecting a non-existent effect).

The questionnaire, developed in mid-2021, did not track how many times respondents contracted COVID-19 or how many doses and types of vaccines they received. These variables inevitably contribute to variability in the results and increase the risk of failing to detect existing associations in statistical tests.

## 5. Conclusions

Our study suggests that contracting COVID-19, especially when accompanied by severe symptoms, may negatively affect health and cognitive performance. It also indicates that even among young individuals, who mostly experienced relatively mild cases of COVID-19, some symptoms (such as frequent fatigue, impaired memory, and certain health issues) could persist for up to four years. Notably, while these symptoms tend to worsen for everyone over the first three years, they may continue to worsen in women even during the fourth year after infection. In contrast, neither participants’ health nor cognition correlated significantly with the time elapsed since vaccination, and nearly all nonsignificant trends (except for the levels of fatigue) were beneficial. SARS-CoV-2 variants differ in the severity and nature of their long-term impacts on health and cognitive performance. The ancestral SARS-CoV-2 variant had the most severe long-term consequences, followed by the Alpha variant, with the Delta variant showing intermediate effects. The Omicron variant had the mildest impact on physical health and cognition, though it may have a relatively stronger effect on mental health. These differences do not appear to be strongly related to variations in the clinical course of the original acute illness or the time elapsed since COVID-19. Individuals who contracted COVID-19 after being vaccinated tend to fare better in terms of health and cognition compared to those who contracted it before vaccination. However, it is important to emphasize that future research controlling for additional factors such as the number of infections, antibody levels, vaccine types, the number of doses received, and probable pre-COVID-19 infection differences among subjects can provide greater insight into these findings.

This study also highlighted the need for caution when interpreting the results of cross-sectional studies. The likelihood of infection and adherence to preventive measures, including vaccination, often correlate with an individual’s overall health status and their perception of the risks associated with the illness. As a result, observational studies may occasionally find that unvaccinated individuals who were infected show better health in certain parameters compared to vaccinated individuals who were not infected. Given that randomized experimental studies are rarely ethically permissible in human medicine to assess the impact of infection or vaccination on health, prospective case–control studies should be prioritized. In observational studies, it is essential to consider the Bradford Hill criteria for causality [44], a set of principles that help assess whether observed associations are likely to be causal [45]. In particular, the dose–response criterion highlights the need to examine correlations between health outcomes and both the severity of illness and the time elapsed since illness or vaccination.

## Figures and Tables

**Figure 1 biomedicines-13-00069-f001:**
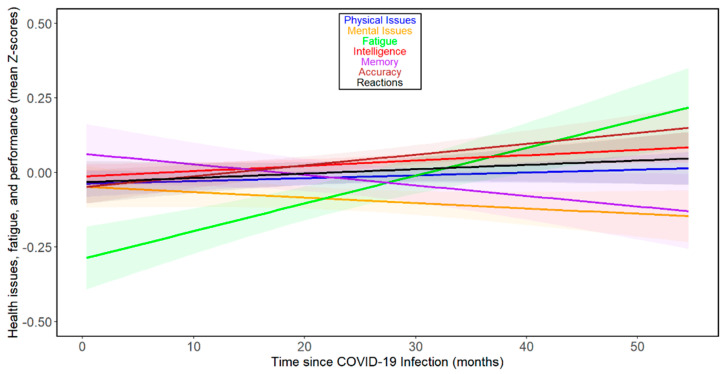
Changes in health- and performance-related variables with time since COVID-19 infection—all participants. Based on the comparison of BIC values, all observed dependencies were best approximated by a first-degree polynomial (a straight line). The shaded areas around the lines indicate 60% confidence intervals. It should be noted that the slopes of the lines on the graph cannot be directly compared with the (more accurate) Tau values calculated using the non-parametric partial Kendall test, which is less sensitive to the presence of outliers and additionally controlled for the influence of 3 confounding variables (sex, age, and year of survey).

**Figure 2 biomedicines-13-00069-f002:**
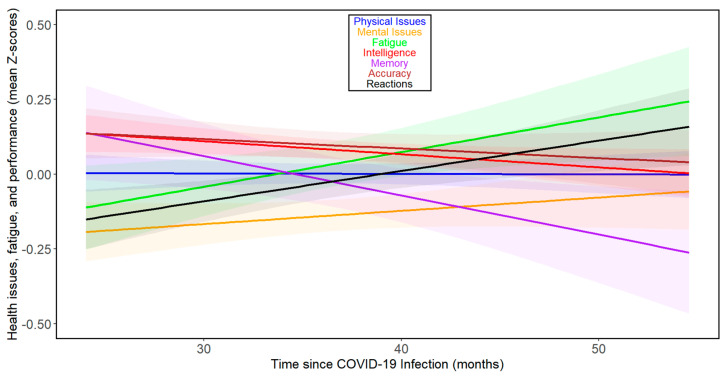
Changes in health- and performance-related variables with time since COVID-19 beyond 24 months post-infection in all participants. For the legend, refer to Figure 1.

**Figure 3 biomedicines-13-00069-f003:**
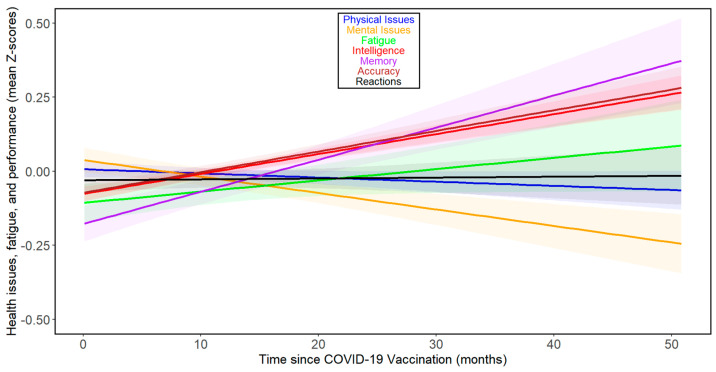
Changes in health- and performance-related variables with time since COVID-19 vaccination in all participants. For the legend, refer to Figure 1.

**Table 1 biomedicines-13-00069-t001:** Descriptive statistics of health and cognitive performance-related indices by sex.

	Mean		*N*		Standard Deviation	
	Women	Men	Women	Men	Women	Men
Physical Health Issues	0.05	−0.13	415	169	0.45	0.39
Mental Health Issues	0.08	−0.20	415	169	0.67	0.60
Fatigue	4.56	4.23	412	166	1.11	1.23
Intelligence	9.28	9.63	405	164	1.47	1.38
Memory	−0.02	−0.30	413	168	1.03	0.95
Reactions	0.00	0.00	415	169	0.65	0.64
Accuracy	−0.03	0.08	415	169	0.57	0.50

*Except for fatigue and intelligence, other indices were calculated as mean Z-scores.*

**Table 2 biomedicines-13-00069-t002:** Correlations between health- and performance-related variables and COVID-19-related variables controlled for age, sex, and survey year (all subjects).

	Infected	Course	Months Since Infection	Vaccination	Months Since Vaccination	COVID-19 After Vaccination	Age	Sex
Physical Health Issues	0.012	** 0.211 **	0.013	** 0.062 **	0.013	−0.012	0.012	** 0.163 **
Mental Health Issues	−0.049	0.066	−0.045	0.048	−0.015	−0.025	0.036	** 0.162 **
Fatigue	−0.042	** 0.093 **	0.044	** 0.089 **	0.036	−0.005	−0.005	** 0.097 **
Intelligence	0.021	−0.057	−0.071	−0.044	0.005	** 0.171 **	−0.021	** −0.099 **
Memory	0.045	−0.010	−0.053	−0.039	−0.011	0.029	**−0.055**	** 0.111 **
Reactions	0.029	0.029	0.036	** −0.075 **	0.027	−0.067	−0.009	0.020
Accuracy	0.001	** −0.082 **	−0.025	−0.015	0.051	** 0.194 **	−0.053	** −0.096 **
				***p*-values**				
Physical Health Issues	0.655	0.000	0.769	0.026	0.707	0.828	0.672	0.000
Mental Health Issues	0.079	0.101	0.298	0.084	0.670	0.653	0.191	0.000
Fatigue	0.128	0.021	0.313	0.002	0.314	0.931	0.846	0.001
Intelligence	0.451	0.161	0.106	0.122	0.887	0.002	0.451	0.000
Memory	0.108	0.809	0.227	0.167	0.757	0.612	0.048	0.000
Reactions	0.302	0.473	0.411	0.007	0.444	0.230	0.744	0.467
Accuracy	0.973	0.041	0.566	0.587	0.145	0.001	0.057	0.001

Significant correlations (Kendall Tau) are presented in bold. The *p*-values that were significant after the application of the Benjamini–Hochberg correction for multiple testing (with FDR set at 0.1) are underlined.

**Table 3 biomedicines-13-00069-t003:** Correlations between health- and performance-related variables and SARS-CoV-2 variants controlled for age, sex, and survey year (all subjects).

	Ancestral	Alpha	Delta	Omicron
Physical Health Issues	0.050	−0.024	0.001	−0.029
Mental Health Issues	−0.051	0.018	−0.023	0.052
Fatigue	** 0.126 **	** −0.103 **	−0.062	−0.007
Intelligence	** −0.099 **	−0.076	0.018	** 0.172 **
Memory	** −0.119 **	−0.035	0.065	0.068
Reactions	** 0.097 **	** −0.095 **	−0.025	−0.024
Accuracy	−0.023	−0.073	0.010	0.075
		***p*-values**		
Physical Health Issues	0.251	0.585	0.988	0.510
Mental Health Issues	0.245	0.675	0.601	0.237
Fatigue	0.004	0.018	0.159	0.881
Intelligence	0.024	0.083	0.676	0.000
Memory	0.007	0.428	0.141	0.119
Reactions	0.027	0.030	0.565	0.584
Accuracy	0.599	0.093	0.812	0.088

Significant correlations (Kendall Tau) are presented in bold. The *p*-values that were significant after the application of the Benjamini–Hochberg correction for multiple testing (with FDR set at 0.1) are underlined.

## Data Availability

The complete dataset is available at Figshare: https://doi.org/10.6084/m9.figshare.27618876, accessed on 27 December 2024.

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
