# Peer review of "Persistent Health and Cognitive Impairments up to Four Years Post-COVID-19 in Young Students: The Impact of Virus Variants and Vaccination Timing"

_biomedicines, 2024, doi:10.3390/biomedicines13010069_

Round 1
Reviewer 1 Report
Comments and Suggestions for Authors
It is a descriptive study with an acceptable sample size but its conclusions go beyond what can be demonstrated by such a study and should be reviewed and avoid being so dogmatic. Conclusions should NOT be so categorical as they are based on a simple survey.
Moreover, the introduction should be shortened. Material and methods should clearly detail the inclusion and exclusion criteria of the respondents and whether they signed written informed consent or not.
On page 11 of the manuscript, lines 405 and 406 there is a different font.
Point 4.3 needs to be rewritten as it is not possible to attribute causality with survey data: regarding the issue of vaccination and severity of COVID19 disease.
With these corrections it could be published.
Author Response
Please see the attachment. Our responses and description of changes are attached as a Word file.

Reviewer 2 Report
Comments and Suggestions for Authors This study investigated the long-term consequences of COVID-19 infection on the health and cognitive performance of young university students. It assesses symptoms related to continuing infections up to four years after infection, in relation to virus variants and vaccination timing. Therefore, such a study will be highly pertinent since it yields substantial information about the long-term physical and cognitive health impacts of COVID-19, particularly in young populations less discussed group. The comments on the authors’ considerations are enumerated below.- Introduction: The introduction is clear and relevant; it provides a good background on the long COVID and possible long-term effects of SARS-CoV-2. This introduction adequately sets the research questions. However, the literature review should have been more extensive, as well as mentioning more recent studies on cognitive and physical health after COVID.
- Methods: While the study design is strong, detailed methodological information regarding non-parametric analysis and multivariate testing is lacking. The statistical test choices should be reported in greater detail, and limitations with respect to the cross-sectional approach should be specified to enhance the reproducibility of the study.
- Results: The results are well presented and have appropriate statistical analyses to support them. It would be very great to see tables and figures used more effectively and, where possible, simplified to improve the clarity. Consider the addition of a summary table that outlines the key findings by variant and health outcomes to improve scan readability.
- Discussion: The discussion is good and interprets the findings, but a little more balance will be welcome. Of course, the authors must discuss some possible confounding factors, such as unmeasured factors that can affect health outcomes. Limitations to name a few, reliance on self-reported data must be discussed to strengthen scientific rigor.
- Major Flaws:
Reviewer 3 Report
Comments and Suggestions for Authors
1. This is a valuable study to investigate the persistent health and cognitive Impairments up to four years post-COVID-19 in young students,and provide information basis for government departments to take effective prevention and control measures.
2. The conclusion in the abstract is not accurate enough. “The study also discusses some limitations inherent in cross-sectional studies, particularly those arising from the stronger tendency of individuals with poorer health, compared to healthier individuals, to avoid infection and prioritize vaccination. To mitigate potential bias, this study proposes focusing on factors such as illness severity and time since infection or vaccination when analyzing persistent symptoms.” This paragraph should not be the main conclusion of the paper.
3. Too many keywords and repetition.
4. The classification of virus variants should be named scientifically. Suggest not using 'Wuhan variant'.
5. Table 1 should provide general information about the research subjects. Why does the author list 'Descriptive statistics of health and cognitive performance related indicators by sex' in Table 1? What is the purpose?The format of Table 1 is not standardized.Table 1 shows that the standard deviation of some indicators' measured values is much larger than the mean. Are the results reliable?
6. The main text lacks data support for gender statistical analysis. Suggest adding figures or tables to sections 3.3.1, 3.3.2, and 3.3.3.
7. Add”Impact of Gender” in the discussion.
8. Suggest streamlining supplementary materials.
Author Response
This is a valuable study to investigate the persistent health and cognitive Impairments up to four years post-COVID-19 in young students,and provide information basis for government departments to take effective prevention and control measures.
Thank you
- The conclusion in the abstract is not accurate enough. “The study also discusses some limitations inherent in cross-sectional studies, particularly those arising from the stronger tendency of individuals with poorer health, compared to healthier individuals, to avoid infection and prioritize vaccination. To mitigate potential bias, this study proposes focusing on factors such as illness severity and time since infection or vaccination when analyzing persistent symptoms.” This paragraph should not be the main conclusion of the paper.
Thank you very much for your insightful comment—you are absolutely correct! The original “Conclusions” were not truly conclusions at all. Instead of summarizing the key findings of our study, they included speculative commentary and promotional remarks about topics discussed elsewhere in the article. We have now replaced this inappropriate section with the following revised text:
“Conclusions: Overall, our results indicate that even in young individuals who predominantly experienced only mild forms of the infection, a gradual decline in health and fitness can occur over a span of four years post-infection. Notably, some negative trends—at least in men—only began to stabilize or even reverse during the fourth year, whereas in women, these trends showed no such improvement. These findings suggest that the long-term public health impacts of COVID-19 may be more severe and affect a much broader population than is commonly assumed.”
We believe this revised conclusion now appropriately reflects the core outcomes of the study. Thank you once again for bringing this to our attention.
- Too many keywords and repetition.
We reduced the number of Keywords by removing those that are already present in the Title of the article.
Originally: SARS-CoV-2; COVID-19; Covid-19 variant; cognition; mental health; physical health; long-term effects; long covid; vaccination
Now: SARS-CoV-2; cognition; mental health; long-term effects; long covid.
- The classification of virus variants should be named scientifically. Suggest not using 'Wuhan variant'.
Thank you for your comment. We agree that scientifically accurate terminology is important. In the revised manuscript, we have replaced the term "Wuhan variant" with "ancestral SARS-CoV-2 variant" and “Ancestral” in tables). This term is commonly used in the scientific literature to describe the original strain of the virus.
- Table 1 should provide general information about the research subjects. Why does the author list 'Descriptive statistics of health and cognitive performance related indicators by sex' in Table 1? What is the purpose?The format of Table 1 is not standardized.Table 1 shows that the standard deviation of some indicators' measured values is much larger than the mean. Are the results reliable?
Thank you for your detailed feedback on Table 1. We appreciate the opportunity to address your concerns and provide additional context.
- Purpose of sex-stratified description in Table 1:
Our previous studies have shown, and the current study confirms, that the long-term consequences of COVID-19 can differ between men and women. Therefore, a sex-stratified description of the data is essential. We have added this justification to the text of the manuscript to clarify the rationale behind this approach:
“In contrast, the long-term impact of COVID-19 on health and performance, particularly the dynamics of these changes over time, differed by sex (see below). Accordingly, all relevant analyses were stratified by sex, and descriptive statistics (Table 1 and Supplementary Table S1) were presented separately for men and women.”
- Table formatting:
The current format of the tables reflects the editorial adjustments made after the submission of the manuscript. We assume that it will be typeset differently in the final publication. - Standard deviations and Z-scores:
Most variables in the table have comparable means and standard deviations because they are calculated as averages of Z-scores, standardized using the formula: Z = (X - mean) / SD. For some variables, larger standard deviations result from the presence of outliers. As our study was both confirmatory and exploratory, we decided to retain the data from outliers to ensure that their information would not be lost. This decision was based on the fact that, although the removal of these outliers would be statistically possible, omitting these data would be conceptually erroneous, as they could provide critical insights into the variability and underlying patterns within the dataset. Outliers often represent significant phenomena or rare events that can inform our understanding of the subject being studied. By excluding them, we would risk losing valuable information that could lead to a more comprehensive analysis and interpretation of the data. Accordingly, to address this issue, we employed nonparametric methods for analyzing correlations, as they are robust to the influence of outliers.
We hope this explanation addresses your concerns.
- The main text lacks data support for gender statistical analysis. Suggest adding figures or tables to sections 3.3.1, 3.3.2, and 3.3.3.
Unfortunately, adding the required data to the main text is technically unfeasible. Each of the three chapters would require a new table with 12 (men, women) or 13 (all subjects) columns and 46 rows. Therefore, we had to move the tables with these important results (effect sizes Tau), along with equally large tables containing corresponding values of statistical significance, to the supplementary materials.
Theoretically, three graphs could be added to the main text, showing the effect of one of the main studied factors—most likely the severity of the disease—on the seven primary outcome variables. However, a graph stratified by sex would be relatively cluttered and would contain the same information, or rather a small and somewhat arbitrarily selected portion of it, that is already fully detailed in the supplementary tables. Including such graphs (or tables) and the necessary accompanying text referring to them would almost double the length of the Results section without increasing the informational value of the article.
The tables in the supplementary materials also illustrate the effect of sex on all studied variables. (In the questionnaire, we asked about both sex assigned at birth and gender—how participants currently identify themselves on a scale from 0: definitely as a woman, to 100: definitely as a man—but for this study, we only included the effect of sex assigned at birth.)
- 7. Add”Impact of Gender” in the discussion.
We added the following subchapter to the discussion:
“We did not observe any differences in the risk of infection or the severity of COVID-19 between men and women. However, men and women differed significantly in physical and mental health, as well as in cognition-related variables. With the exception of memory and reaction times, women scored significantly worse on all seven focal variables (see Supplementary Table S7). This was the primary reason for controlling for sex in the confirmatory part of the study and for analyzing the effects of COVID-19-related variables on health and cognition separately for men and women in the main exploratory analysis. This approach, however, did not allow for formal testing of the effect of the interaction between sex and COVID-19-related variables on health and performance. Nevertheless, visual inspection of Supplementary Figures S1 and S2 suggests notable trends. Among individuals who had recovered from COVID-19, men reported significantly better physical and mental health immediately after infection. Over time, however, these measures showed substantial declines. In contrast, women's physical health remained relatively stable throughout the observation period, while their mental health gradually improved.
Men reported lower fatigue levels than women shortly after infection, but fatigue in men progressively worsened over time. In women, fatigue levels also worsened, but at a faster rate, resulting in a pronounced difference between sexes by the end of the study period. Intelligence in men was slightly higher shortly after infection but gradually declined, while intelligence in women showed a slight increase over time, eventually surpassing that of men by the end of the observation period.
Memory was markedly poorer in men immediately after infection and declined at a much faster rate, leading to a difference between men and women that was twice as large at the end of the study period compared to its beginning. In contrast, accuracy in cognitive performance tests was slightly higher in men immediately post-infection and increased markedly more over time than in women, leading to a notable advantage for men by the end of the study. Reaction times were initially slightly better in men and gradually worsened at a similar rate to those in women.
The most pronounced differences emerged during the fourth year post-infection. While women continued to experience declines in health and cognitive performance (except for memory and accuracy in cognitive tests), men showed improvement in all parameters except reaction times. However, it is important to note that the dataset included only 15 men who had been infected more than three years prior, which limits the generalizability of these observations.”
- Suggest streamlining supplementary materials.
Thank you for this suggestion. To facilitate better navigation through the extensive material, we have added a table of contents with hyperlinks to individual tables at the beginning. Additionally, we have made an effort to improve the layout and organization of the tables and figures overall.
Reviewer 4 Report
Comments and Suggestions for Authors
General feed back
This cross -sectional study was conducted online on 584 students (415 females vs. 169 females) from Czech Republic (mean age 21.5 years ) enrolled from undergraduate basic evolutionary biology courses, to investigate the impact of virus variants, vaccination, disease seveity, and time since infection on cognitive post -COVID-19 conditions.
Specific comments
· In general, the manuscript is excessively long and dispersive and would benefit from some significant reduction (in all parts).
· For instance, the introduction is too long and disjoint. Too many studies are unnecessarily reported, without an apparent criterium. I recommend to summarize this part narrowing down the main message (e.g. most prevalent symptoms, cognitive impairments and most relevant risk factors - e.g. severity of COVID-19 and prolonged viral persistence), focusing on cognitive effects in young individuals, hence significantly reducing lines 31-80.
· Abstract (lines 23-27): These are limitations and should be reported in the full manuscript, not in abstract conclusions. Conclusions of an abstract should just succinctly comment the study findings.
· Line 46-57: please provide the time periods of these studies, as this information reflects the circulating variants.
· Line 48: change “generalized” to “aspecific”
· Line 48-49: “muscle weakness” and “myalgia” have the same meaning
· Line 50: “with 15.6% of it representing cognitive impairment”.. awkward expression, to be rephrased
· Line 52: “in this regard” to be removed
· Line 59-60: severity od COVID-19, early pandemic waves and prolonged viral shedding time are reportedly relevant risk factors for long-COVID-19 [recommended citation: PMID: 38140174]
· Line 67: “Britian”, to be corrected in “Britain”
· Line 94: “the rates of error”… what “error”?
· Line 106-108: this is critical point, since post-COVID-19 symptoms (including work ability) are reportedly more likely in patients with pre-existing mood disorders depression/anxiety [recommended additional citation: PMID 36676046]
· Line 117-138: The lines are fully unnecessary and be expressed in 1-2 lines. Moreover, the setting and timeline should be mentioned
· Line 143-148: this information is obvious and redundant and should be removed.
· Line 153-159: it is misleading for the purpose of the current study to report information on the exam students had to take and survey on mood, optimism, etc. This study should focus on long-COVID-19 investigation.
· Line 167-177: likewise, these lines are useless for the current study and should be definitely removed. This study should focus on health status and COVID-19 related information.
· Line 172: “are available in [13]”… change to “are available elsewhere [13]”
· 193-94: self-reported blood pressure is rather vague…. What does “very low”, “rather low” etc. mean?
· Fatigue, the most frequent post_COVID-19 symptoms is rather aspecific and should be screened (recommended citations: [PMID: 38140174; PMID: 34631916])
· Among post-COVID-19 conditions, heart rate variability should be mentioned [recommended citation PMID: 38795941]
· 2 frequently reported long-COVID-19 symptoms were not considered: concentration deficit and insomnia
· Line 242: “which is the non-parametric equivalent of the independent t-test,”, this sentence is obvious and can be removed. No need to define Wilcoxon non-parametric test
· Line 254-264: these lines are unnecessary and can removed for simplicity
· Line 277-278: “who had contracted 277 COVID-19, 29 (29.0%) contracted the virus after”… “contracted” repeated twice, please find a synonym
· Line 287: change “men and women” to “males and females”
· Lines 294-306: These are methods and should not be here. Moreover, this is not a statistical manual, hence this obvious didactical description should be removed. Presenting the results pertaining to those hypotheses is enough
· Line 314-315: “The COVID-19 vaccine exhibited significant positive associations with physical health issues and fatigue and a negative association with reaction times”… this is unclear. This sentence should clearly and briefly report whether COVID-19 vaccination improved or worsened health status of COVID.19 patients.
· Line 315-316: this is a striking finding
· Line 331: “The results indicated that…” can be removed so that the sentence starts with “There were no significant associations…etc.”
· The results section is far too long, dispersive, unnecessarily detailed and difficult to follow. It should definitely be halved (at least)
· Line 262-264: unnecessary lines to be removed from results section
· Table 2: why not fitting a multiple linear regression model instead ?
· In results section it would be useful to have some statistical estimates reported
· It is unclear whether the authors took into account also the effect of reinfections and hybrid immunity conferred by previous infection and vaccination.
· Line 457-460: these lines can be removed
· Line 522-523: it is unclear from this sentence whether level of depression decreased or increased with time since COVID.19 vaccination.
· Line 542-557: these are study strengths and should be moved to the section strength and limitations. The first part of discussion should summarize the main findings
· Line 579-581: these are just speculations. Severe COVID-19 in children is reportedly negligible.
· Line 621-22: There is also evidence that vaccination and reinfection decreased the risk of long-COVID-19 syndrome (recommended citation: PMID: 38140174]. The same study reported Wuhan/Alpha variant and severe COVID-19 consistent risk factors for long-COVID-19
· Line 644: “suggesting a lack of causal relationship”, this is a speculation not supported by evidence
· Line 709-711: these lines are unnecessary. This is a scientific article, not a statistical manual.
· Line 730-742: The lines (which are also excessively too long to read) are study limitations, should be removed from conclusions.
· Line 713-728: how could this study establish that any persisting poor cognitive performance is linked to COVID-19?
· Table 2: these correlation coefficients should be controlled also for the effect of vaccination status (number of doses of COVID-19 vaccine) plus any co-morbidities
Comments on the Quality of English LanguageAn overall revision is recommended.
But the most important point is the quality of presentation, since the manuscript is far too dispersive and readers can get lost
Author Response
Referee 4
This cross -sectional study was conducted online on 584 students (415 females vs. 169 females) from Czech Republic (mean age 21.5 years ) enrolled from undergraduate basic evolutionary biology courses, to investigate the impact of virus variants, vaccination, disease seveity, and time since infection on cognitive post -COVID-19 conditions.
Specific comments
- 9. In general, the manuscript is excessively long and dispersive and would benefit from some significant reduction (in all parts).
- For instance, the introduction is too long and disjoint. Too many studies are unnecessarily reported, without an apparent criterium. I recommend to summarize this part narrowing down the main message (e.g. most prevalent symptoms, cognitive impairments and most relevant risk factors - e.g. severity of COVID-19 and prolonged viral persistence), focusing on cognitive effects in young individuals, hence significantly reducing lines 31-80.
We agree that the manuscript is long and complex. However, there are objective reasons for this. The design of the present study— a serial cross-sectional study—essentially represents three independent cross-sectional studies conducted over three consecutive years. Moreover, the spectrum of assessed health-related variables is very broad, and the study also included experimental testing of cognitive performance (IQ, memory, ability to concentrate, reaction times) using a panel of cognitive tests. In total, the manuscript reports the results of approximately 1,700 computationally intensive non-parametric multivariate statistical tests, most of which are presented in the supplementary tables.
It would certainly be possible to split the manuscript into several shorter articles. However, doing so would result in a total length far exceeding the current manuscript and would ultimately provide a less coherent view of the long-term processes affecting young individuals who have experienced COVID-19. In its original form (before incorporating new sections required by four reviewers), the manuscript contains approximately 8,300 words excluding tables, figures, the reference list, and the abstract. Compared to similar articles in the literature, this length is not particularly unusual for articles covering complex topics such as ours.
The introduction alone consisted of 1,277 words, which is entirely standard for articles addressing such a complex issue and having such a broad scope, as observed in the related literature. It should also be noted that, for example, Reviewer 1 explicitly requested an expansion of the introduction, and all four reviewers (including Reviewer 4) requested the inclusion of additional sections or citations in the introduction and discussion. Organically incorporating these new elements inevitably resulted in further lengthening of the text. Nevertheless, we have made every effort to ensure that it is as precise and concise as possible without omitting necessary information or including excessive details.
We agree with the reviewers that it would be ideal for the manuscript to be significantly shorter while also being more comprehensive in content. However, considering the contradictory practical implications of these suggestions—namely, the need for both brevity and greater detail—along with the concerns regarding the scope and extent of our study, we find that these requirements are not manageable.
- 10. Abstract (lines 23-27): These are limitations and should be reported in the full manuscript, not in abstract conclusions. Conclusions of an abstract should just succinctly comment the study findings.
Thank you for bringing this to our attention. We have now replaced this inappropriate section with the following revised text:
“Conclusions: Overall, our results indicate that even in young individuals who predominantly experienced only mild forms of the infection, a gradual decline in health and fitness can occur over a span of four years post-infection. Notably, some negative trends—at least in men—only began to stabilize or even reverse during the fourth year. These findings suggest that the long-term public health impacts of COVID-19 may be more severe and affect a much broader population than is commonly assumed.”
We believe this revised conclusion now appropriately reflects the core outcomes of the study.
- 11. Line 46-57: please provide the time periods of these studies, as this information reflects the circulating variants.
Thank you for the suggestion, the information is provided now.
- 12. Line 48: change “generalized” to “aspecific”
Done.
13. Line 48-49: “muscle weakness” and “myalgia” have the same meaning
Corrected, thank you.
- 14. Line 50: “with 15.6% of it representing cognitive impairment”.. awkward expression, to be rephrased
We changed:
“The most commonly reported symptoms were generalized (fatigue, muscle weakness, joint pain, hair loss, sweating, myalgia, skin rash, and chill) (60.7%), mental (48.3%), cardio-pulmonary (39.8%), neurological (37.1%, with 15.6% of it representing cognitive impairment), and digestive (19.1%) symptoms [7].”
To
"The most commonly reported symptoms were aspecific (fatigue, muscle weakness, joint pain, hair loss, sweating, myalgia, skin rash, and chill) (60.7%), mental (48.3%), cardio-pulmonary (39.8%), neurological (37.1%, including cognitive impairment in 15.6% of cases), and digestive (19.1%) symptoms [7]."
- 15. Line 52: “in this regard” to be removed
Done.
- 16. Line 59-60: severity od COVID-19, early pandemic waves and prolonged viral shedding time are reportedly relevant risk factors for long-COVID-19 [recommended citation: PMID: 38140174]
Thank you for bringing this study to our attention. We have included the following paragraph in the first paragraph of the Introduction:
“Similarly, a recent study of healthcare workers in Italy identified severity of acute COVID-19, early pandemic waves (ancestral SARS-CoV-2 and Alpha variants), and prolonged viral shedding time as significant risk factors for long COVID-19. This study also highlighted that the prevalence of long COVID-19 decreased with subsequent infections and was lower during the Omicron wave, likely due to increased immunity and milder strains [10].”
- 17. Line 67: “Britian”, to be corrected in “Britain”
Corrected.
- 18. Line 94: “the rates of error”… what “error”?
Thank you for pointing this out. We have clarified the text to specify the type of errors being referred to. The revised sentence now reads:
"In the first 24 months following the infection, improvements in both physical and mental well-being were observed, along with a reduction in the frequency of errors in attention-demanding tests."
- 19. Line 106-108: this is critical point, since post-COVID-19 symptoms (including work ability) are reportedly more likely in patients with pre-existing mood disorders depression/anxiety [recommended additional citation: PMID 36676046]
Thank you for pointing out the relevance of pre-existing psychological conditions as risk factors for long COVID-19 symptoms and for recommending the citation (PMID: 36676046). We have incorporated this into the revised manuscript as follows:
"The meta-analysis concluded that many children and young people experience persistent symptoms after SARS-CoV-2 infection [14]. Nonetheless, it should be noted that some studies suggest that ongoing symptoms and impairments in post-COVID-19 conditions may be influenced by factors beyond the virus itself, particularly psychosocial aspects pre-existing psychological conditions [15]. For example, a study [16] highlighted that pre-existing psychological conditions, such as depression and anxiety, are significant risk factors for persistent symptoms, including psychiatric and neurological complaints, as well as reduced work ability long after COVID-19 recovery."
This addition provides additional context and aligns with the points raised in your comment. We appreciate your suggestion and have revised the text accordingly.
- 20. Line 117-138: The lines are fully unnecessary and be expressed in 1-2 lines. Moreover, the setting and timeline should be mentioned
Thank you for your feedback regarding Lines 117-138. While we understand your concern about brevity, we respectfully disagree with the suggestion to condense this section into 1-2 lines. The aims and scope of the study are a crucial part of the Introduction, as they help readers orient themselves within the study and understand its significance. None of the other three reviewers suggested omitting or drastically shortening this important section.
That said, we have made an effort to streamline this part while retaining its essential content. Additionally, we have clarified the specific timeline for the sub-studies (conducted during the second week of January in 2022, 2023, and 2024) in the Methods section, as suggested.
We hope these adjustments address your concern while maintaining the clarity and comprehensiveness of the Introduction.
- 21. Line 143-148: this information is obvious and redundant and should be removed.
Thank you for your suggestion. However, we believe it is important to retain this section in the methodology for several reasons:
- Modern standards require detailed methodology, including addressing ethical considerations. It is essential and, in most biomedical journals, obligatory to explicitly state how participants were informed about the voluntary nature of their participation, their ability to withdraw at any time, and the confidentiality of their data. This ensures transparency and demonstrates adherence to ethical research practices, which are critical for any study involving human participants.
- In our specific case, this information also addresses concerns about selection bias. By explaining that participants were not informed beforehand that COVID-19 would be a focus of the study, we ensure readers understand that the recruitment process did not predispose participants based on their COVID-19 experience or related interests. This is a crucial point for interpreting the validity of the study’s findings.
We believe that including this information enhances the methodological rigor of the manuscript and provides readers with confidence in the study’s ethical and procedural soundness.
- 22. Line 153-159:it is misleading for the purpose of the current study to report information on the exam students had to take and survey on mood, optimism, etc. This study should focus on long-COVID-19 investigation.
We believe that the information about the preregistration of the study and questionnaire is very important for readers, and removing it would reduce the credibility of the results. Nevertheless, we have decided to comply with the referee’s suggestion and have removed this paragraph.
- 23. Line 167-177: likewise, these lines are useless for the current study and should be definitely removed. This study should focus on health status and COVID-19 related information.
We believe there may have been a misunderstanding. Our study is dedicated to investigating the impact of COVID-19 on health and cognitive performance, and the mentioned section of the methodology describes the tests used to measure cognitive performance. Without this specification, half of the results would lack context and would not make sense to the reader.
- 24. Line 172:“are available in [13]”… change to “are available elsewhere [13]”
Corrected.
- 25. 193-94: self-reported blood pressure is rather vague…. What does “very low”, “rather low” etc. mean?
We agree. This question is not appropriate, and in future studies, we will replace it with a question asking respondents to specify their usual blood pressure values, or we will omit it entirely.
- 26. Fatigue, the most frequent post_COVID-19 symptoms is rather aspecific and should be screened (recommended citations: [PMID: 38140174; PMID: 34631916])
Now, we cited this review in the Introduction.
- 27. Among post-COVID-19 conditions, heart rate variability should be mentioned [recommended citation PMID: 38795941]
Thank you for the suggestion regarding heart rate variability as a post-COVID-19 condition. We did consider incorporating the recommended reference ("Rate variability modulation through slow-paced breathing in health care workers with long COVID: a case-control study," PMID: 38795941), which indeed describes an intriguing intervention technique. However, we were unable to find a suitable way to integrate this reference into the current manuscript without deviating from its primary focus and scope. We appreciate your understanding.
- 28. 2 frequently reported long-COVID-19 symptoms were not considered: concentration deficit and insomnia
We agree—it is indeed unfortunate that these symptoms were not considered. However, the study was initiated in early January 2022, with the questionnaire designed during the summer of 2021 and the protocol approved by the IRB in the fall of 2021. At that time, the (published) knowledge regarding long COVID was significantly more limited than it is today. To provide readers with a clearer understanding of when the study was conducted, we have updated the Materials and Methods section in the revised version, replacing paragraph
“Over three consecutive years (2022, 2023, and 2024, all students enrolled in the basic course in evolutionary biology were invited to take part in an anonymous online study.”
with
“Over three consecutive years (2022, 2023, and 2024), always in the second week of January, all students enrolled in the basic course in evolutionary biology were invited to take part in an anonymous online study.”
- 29. Line 242: “which is the non-parametric equivalent of the independent t-test,”, this sentence is obvious and can be removed. No need to define Wilcoxon non-parametric test
Done.
- 30. Line 254-264: these lines are unnecessary and can removed for simplicity
We respectfully disagree with the suggestion to remove these lines. This information is crucial for characterizing the dataset used in the study and for evaluating the justification and potential impact of excluding participants with probable but unconfirmed COVID-19. Additionally, it provides valuable context about the epidemiological situation during the study period. We believe that retaining this section enhances the transparency and robustness of the study's methodology and findings.
- 31. Line 277-278:“who had contracted 277 COVID-19, 29 (29.0%) contracted the virus after”… “contracted” repeated twice, please find a synonym
Thank you. We have revised the sentence to:
“Of the 100 vaccinated women who had COVID-19, 29 (29.0%) contracted the virus after their last vaccination; among 49 men, this was the case for 17 (34.7%) (Chi2(1) = 0.499, p = 0.479).”
- 32. Line 287:change “men and women” to “males and females”
Thank you for your suggestion. However, we respectfully disagree with changing "men and women" to "males and females." Referring to study participants as "males" and "females" strikes us as somewhat inappropriate in this context. Throughout the manuscript, we have deliberately used "men and women," or occasionally "male or female students (respondents)," to maintain a respectful and professional tone.
- 33. Lines 294-306: These are methods and should not be here. Moreover, this is not a statistical manual, hence this obvious didactical description should be removed. Presenting the results pertaining to those hypotheses is enough
Thank you for your thoughtful comment. However, we believe that this section should remain in the manuscript for several important reasons. First, explicitly stating the hypotheses provides a clear framework for understanding the confirmatory part of the study, which significantly enhances the readability and coherence of what is otherwise a complex article. It helps the reader quickly identify the key research questions and the corresponding results, making the study more accessible and easier to follow.
Second, presenting the hypotheses in this way is a standard practice in contemporary biomedical literature. It is particularly useful in studies that combine exploratory and confirmatory approaches, as it delineates which findings address pre-specified hypotheses and which are derived from post-hoc analyses. This approach adds clarity and structure to the manuscript, ensuring that readers can distinguish between hypothesis-driven results and exploratory findings.
Finally, we feel that the description of the statistical approach is essential for transparency. While it is concise, it ensures that the methods used to analyze the data are directly linked to the hypotheses being tested, which is critical for reproducibility and for readers seeking to critically evaluate the study.
We hope these points clarify the importance of retaining this section and its role in enhancing the manuscript’s clarity and rigor. Thank you again for raising this point, and we appreciate your understanding.
- 34. Line 314-315: “The COVID-19 vaccine exhibited significant positive associations with physical health issues and fatigue and a negative association with reaction times”… this is unclear. This sentence should clearly and briefly report whether COVID-19 vaccination improved or worsened health status of COVID.19 patients.
Thank you for your feedback. We carefully reviewed the sentence and understand the importance of absolute clarity. We have rephrased it to ensure that the findings are explicitly stated:
"COVID-19 vaccination was associated with poorer physical health and increased fatigue but with improved (shorter) reaction times in vaccinated students compared to unvaccinated ones."
It is important to note that the results presented later in the manuscript, and discussed in the context of other published studies in the chapter Discussion, indicate that vaccination does not worsen health or increase fatigue. The observed association likely arises because individuals at higher risk—primarily those with poorer health—are more likely to prioritize vaccination.
- 35. Line 315-316: this is a striking finding
Thank you for highlighting this point. While it may initially appear to be a striking finding, we believe it is not particularly surprising, as we discuss in detail later in the manuscript. There is substantial evidence in the literature, including findings from some of our previous studies, demonstrating that individuals in poorer health tend to prioritize vaccination earlier and more frequently than healthier individuals. This behavior creates the observed negative association between health and vaccination, which is a well-documented phenomenon in similar contexts.
- 36. Line 331: “The results indicated that…” can be removed so that the sentence starts with “There were no significant associations…etc.”
Done.
- 37. The results section is far too long, dispersive, unnecessarily detailed and difficult to follow. It should definitely be halved (at least)
As we already explained in response to point 9:
- It is not feasible to present this extensive study more economically. The design—a serial cross-sectional study conducted over three consecutive years—provides a comprehensive analysis through multiple phases of data collection, which is literally tantamount to three independent studies. It also includes a wide range of health-related variables and experimental testing of cognitive abilities. The results are based on approximately 1,700 computationally intensive non-parametric statistical tests, most of which are presented in supplementary tables to ensure transparency.
- The length of the study is not unusual. At approximately 8,300 words (excluding tables, references, and the abstract), the manuscript aligns with the typical length of articles addressing such complex topics. Shortening or splitting the study into multiple articles would result in a greater total length and a loss of coherence in presenting the findings. Accordingly, we have meticulously balanced the content and its length to ensure that neither was compromised at the expense of the other.
In fine, we believe the manuscript already represents a reasonable compromise between detail and readability, given the scope and complexity of the study.
- Line 262-264: unnecessary lines to be removed from results section
While we appreciate your concern regarding brevity, we believe it is essential to provide this information to enhance the reader’s understanding of the final dataset’s texture, as is commonly practiced in biomedical literature. We believe that removing it from the text will cause more harm than good.
- 39. Table 2: why not fitting a multiple linear regression model instead ?
There are many approaches to analyzing the same dataset. We chose a method based on partial Kendall regression, a nonparametric multivariate technique that, due to its computational complexity and the historical lack of suitable software, has only recently begun to be widely applied. This is a highly robust method that does not require specific distributions for the variables and is unaffected by very uneven group sizes (in our case, only 6% of students fell into the unvaccinated category). Another significant advantage for our study was that this method could be uniformly applied to continuous, ordinal, and binary data. This consistency simplified the analysis, made the manuscript more concise, and improved its clarity, resulting in a more reader-friendly presentation.
In addition, we conducted a number simple linear regression analyses solely to examine the trajectory of the index variables over time since infection. Therefore, employing multiple linear regression would not be feasible for our purpose.
- In results section it would be useful to have some statistical estimates reported
We are unsure what specific additional information the referee is suggesting. For all the tests performed, we provide not only significance levels but also all necessary results describing the respective effects, both in the text and in the tables. In the case of the partial Kendall test, the Kendall Tau values already convey information about both the direction and the effect size of the respective associations. If the referee feels that any specific type of estimate is still missing, we would appreciate further clarification.
- It is unclear whether the authors took into account also the effect of reinfections and hybrid immunity conferred by previous infection and vaccination.
Although this is an interesting point and is surely worth studying, we unfortunately do not have the relevant data available. In mid-2021, when the protocol was being prepared, the questions of repeated vaccination and hybrid immunity were not yet widely considered. In the revised version of the manuscript, we have added the following paragraph to the Strengths and Limitations section:
"The questionnaire, developed in mid-2021, did not track how many times respondents contracted COVID-19 or how many doses and types of vaccines they received. These variables inevitably contribute to variability in the results and increase the risk of failing to detect existing associations in statistical tests."
- 42. Line 457-460: these lines can be removed
This information is essential and must be included either in the Methods section or directly in the Results section, where the outcomes of the corresponding analyses are presented. We have decided to retain its current placement in the Results section because removing it would make it unclear to the reader how the effects of individual virus variants were tested and what the results represent. Keeping this (short) explanation in the Results section ensures clarity and allows the findings to be properly understood and interpreted.
- Line 522-523: it is unclear from this sentence whether level of depression decreased or increased with time since COVID.19 vaccination.
Thank you for the comment. To clarify, we have revised the sentence to ensure it is clear that levels of depression and mental distress decreased with time since vaccination:
"The time elapsed since infection did not show any significant correlations with the output variables, but the time since vaccination was associated with reduced levels of depression and lower mental distress at the time of the survey."
This revised phrasing explicitly states the direction of the association and hopefully should resolve any ambiguity.
- 44. Line 542-557: these are study strengths and should be moved to the section strength and limitations. The first part of discussion should summarize the main findings
We agree that in many journals it is customary and practical to begin the discussion with a brief summary of the study's findings. At the same time, in many journals, this practice is explicitly prohibited in the instructions for authors. Although we almost always include such a summary at the beginning of the discussion, we decided not to do so in this particular case.
Our study addresses so many areas that a concise summary would either be too lengthy or lack the necessary depth, unnecessarily extending the manuscript. Instead, we opted to summarize and discuss individual sets of results in their respective sections of the chapter Discussion.
In the first paragraph of the Discussion, we focus on the differences between this study and a previous one from our laboratory. We also explain the reasons for conducting the current study and its specific contributions to existing knowledge. We believe that for many readers, these questions are likely key, and addressing them upfront allows for a better understanding of the remaining text.
- 45. Line 579-581: these are just speculations. Severe COVID-19 in children is reportedly negligible.
We respectfully disagree that this is merely speculation. Recent studies are beginning to emerge in the literature that support and explicitly discuss this concerning interpretation (e.g., (Wu and Liu, 2023, p. 10). It is important to emphasize that most existing knowledge about the risk factors for severe COVID-19 pertains to the acute phase of the disease. Understandably, we know much less about its long-term consequences due to technical and temporal constraints. Notably, our findings indicate that the virus variants with the most severe long-term effects may not necessarily be those causing the most severe acute cases of COVID-19. It is therefore quite reasonable to assume that the risk and protective factors for the immediate manifestations of COVID-19 and its long-term health consequences may differ from one another.
If SARS-CoV-2 affects the brain, it is reasonable to hypothesize that the developing brain in younger individuals could be particularly vulnerable to these effects. When we wrote the previous version of the manuscript, we did not have additional data to support this possibility and presented it as one of two alternative explanations for the observed correlation between the severity of long-term consequences and the time since infection. However, recent studies now provide evidence supporting this hypothesis, and we have included relevant citations in the revised manuscript to strengthen this point.
- Choudhury et al. Neurologic Manifestations of Long COVID Disproportionately Affect Young and Middle-Age Adults. Annals of Neurology Vol. n/a Issue n/a,
DOI: https://doi.org/10.1002/ana.27128
- Wu and Liu Association of Cognitive Deficits with Sociodemographic Characteristics among Adults with Post-COVID Conditions: Findings from the United States Household Pulse Survey medRxiv 2023 Pages 2023.09.22.23295981 DOI: 10.1101/2023.09.22.23295981
- 46. Line 621-22: There is also evidence that vaccination and reinfection decreased the risk of long-COVID-19 syndrome (recommended citation: PMID: 38140174]. The same study reported Wuhan/Alpha variant and severe COVID-19 consistent risk factors for long-COVID-19
We have added the following information to the Discussion section of the manuscript:
“Similar to the study by Cegolon et al. (2023), our study also demonstrated that the ancestral SARS-CoV-2 variant had the most severe long-term consequences, while the Omicron variant had the mildest.”
- 47. Line 644: “suggesting a lack of causal relationship”, this is a speculation not supported by evidence
Although we still believe that this result, along with other findings presented in this study and elsewhere, supports (but of course does not prove) the hypothesis that vaccination is not the cause of poorer health in vaccinated individuals, we have toned down our statement. It now reads:
"Vaccinated individuals generally exhibited poorer health compared to unvaccinated individuals. However, the difference in their health and performance did not change significantly with time since vaccination, which does not support the existence of a causal relationship between vaccination and poorer health."
- 48. Line 709-711: these lines are unnecessary. This is a scientific article, not a statistical manual.
We have deleted these lines.
- 49. Line 730-742: The lines (which are also excessively too long to read) are study limitations, should be removed from conclusions.
We have edited the paragraph and divided the overly long sentences to improve readability. However, we cannot remove this section, as it represents one of the two most critical conclusions of this study. Without it, the results could be misused by disinformation actors who, contrary to the scientific consensus supported by indisputable data, continue to spread falsehoods claiming that vaccines are harmful and that COVID-19 is essentially harmless for young individuals.
This paragraph is essential for ensuring that our findings are interpreted correctly and in line with the broader scientific evidence. It also highlights the methodological challenges and the importance of applying robust criteria, such as the Bradford Hill criteria, to observational studies in this area. For these reasons, we believe it is crucial to retain this section in the Conclusions.
- 50. Line 713-728: how could this study establish that any persisting poor cognitive performance is linked to COVID-19?
This crucial question is directly addressed in the following paragraph. In brief: In epidemiology and biology, Bradford Hill’s criteria for causality (Hill, 1965) have been widely accepted and applied for over 50 years. These criteria consist of nine principles used to evaluate whether observed associations are likely to be causal (Fedak, Bernal, Capshaw, & Gross, 2015).
In our study, we specifically tested the validity of the first criterion, the dose-response criterion, in two ways. This criterion suggests that A (e.g., contracting COVID-19) is a cause of B (e.g., increased fatigue) if increasing the intensity of A increases the intensity or likelihood of B. First, we tested whether a more severe course of COVID-19 was associated with more frequent and/or more severe health consequences. Second, we tested whether the intensity of health and cognitive issues among individuals who had COVID-19 changed with time since infection.
Both tests yielded very clear results. In both cases, there was a positive correlation between the observed variables, supporting the conclusion that COVID-19 is a cause of the observed changes. This contrasts with the results of our analysis regarding vaccination, where we found no correlation between the intensity of health problems and the time elapsed since vaccination. This lack of association supports the conclusion that vaccination is not likely the cause of the observed health issues.
- 51. Table 2: these correlation coefficients should be controlled also for the effect of vaccination status (number of doses of COVID-19 vaccine) plus any co-morbidities
Although the results would be more appropriate if this suggestion could be employed, as we already explained in point 21, this study was designed and preregistered in mid-2021, at which time the number of COVID-19 infections, as well as the number and types of vaccine doses, were not tracked. Consequently, we do not have the corresponding data available. Similarly, data on potential comorbidities in 21-year-old students were not collected as part of this study.
Comments on the Quality of English Language
An overall revision is recommended.
- But the most important point is the quality of presentation, since the manuscript is far too dispersive and readers can get lost
We appreciate the referee's recommendation for an overall revision and their concern about the quality of the presentation. However, we respectfully disagree with the assessment that the manuscript is too dispersive or difficult to follow. We, along with the three other referees, find that the manuscript is clear and well-structured, given the complexity and breadth of the study.
While the study addresses multiple areas, the text has been meticulously organized into distinct, self-contained sections. This compartmentalization was based on our hypotheses and variables, ensuring both readability and a logical flow. Detailed supplementary materials were included to provide transparency and allow readers to explore specific aspects of the analysis without overloading the main text.
We believe the current structure strikes an appropriate balance between detail and accessibility, allowing the manuscript to be both comprehensive and reader-friendly. We have carefully considered the organization and presentation to ensure clarity while addressing the study's complexity.
Round 2
Reviewer 1 Report
Comments and Suggestions for Authors
The paper is ready for publication.
Author Response
Thank you for your positive evaluation of our manuscript and for the time you dedicated to reviewing our article.
Reviewer 2 Report
Comments and Suggestions for Authors
The authors amended the manuscript greatly
Author Response

(The authors gave the same response as above.)

Reviewer 3 Report
Comments and Suggestions for Authors
The author has made good revisions to the paper according to the reviewer's comments and agrees to accept it.
Author Response

(The authors gave the same response as above.)

Reviewer 4 Report
Comments and Suggestions for Authors
General feed back
· The authors “respectfully” failed to address most recommendations, especially reducing the length of the manuscript or removing redundant or unnecessary sections. I believe the authors struggled in general to express concepts in a clear, simple and concise fashion. There is no contradiction in asking to add some extra-references or provide some more relevant details in some parts, with reducing the length of a manuscript, which is far too long and dispersive and difficult to follow. A lot of information is redundant or unnecessary and could be easily removed or placed in Supplementary files.
· The introduction (almost 3 pages!) is just a serial, hectic and disjoint list of studies selected without an apparent criterium and more suitable for a discussion of scoping review article than for a research paper.
· This is a study on just 584 individuals, of whom just 4 (=1.4%) hospitalized, with results not enough controlled for residual confounders (just sex, age and survey year). The authors stressed on the cross-sectional design as the main limitation, but this is not the main weakness of this study. The main limitation is the poor control for residual confounders (just age. Sex and survey year). Controlling for higher number of potential confounders would definitely help ruling out some spurious associations.
· The authors mentioned some Hills criteria for causality, but biological plausibility is mainly founded on hypotheses and speculations and reproducibility of results may not be feasible yet biased by limited generalizability.
· Methods The FDR set at 10% for Benjamini Hichberg is rather loose criterium. It should be set at 5%
· I do not believe COVID-19 is a problem in children and young individuals, unless developing severe disease. The authors acknowledge this point, but than they affirm that long-COVID-19 is independent from severity of disease. For instance, just 183 children aged 1-17 died for COVID.19 in the entire USA (a country of almost 350 million inhabitants) during 2020-2022, 70% (2 thirds) having some pre-existing medical conditions. (PMID: 39484882). This is a clear indication on the impact of COVID-19 in children.
· Line 571-572: the lower use of medications in males is consistent with the literature, reporting higher prevalence of long-COVID-19 among females, especially with pre-existing depression, suggesting also a possible psychological yet hypochondriac underlying component [e.g. PMID: 36676046).
· 674-679 and Line 735-736: this is the more appropriate conclusions in the entire study, which is in contrast with comments on lines 870-871 (in conclusions)
· Line 713-714: this is another key point, I,e, limited generalizability
· Line 717-718: vaccination has a 95% efficacy against development of severe disease, which is the key risk factor for long-COVID-19 ! this point again questions the generalizability of results, which may be biased by unmeasured factors. In fact, vaccination before COVID-19 was associated with higher cognitive performance compared to vaccination after COVID-19 in this study.
· Line 750-751: I believe the most important factor to explain such results may be poor control for residual confounders.
· In conclusions, the authors should find a way to present their results in a much more concise manner, playing also down their findings in the light of the poor control for residual confounders.
Comments on the Quality of English LanguageCould be improved (see report)
Author Response
The authors “respectfully” failed to address most recommendations, especially reducing the length of the manuscript or removing redundant or unnecessary sections. I believe the authors struggled in general to express concepts in a clear, simple and concise fashion. There is no contradiction in asking to add some extra-references or provide some more relevant details in some parts, with reducing the length of a manuscript, which is far too long and dispersive and difficult to follow. A lot of information is redundant or unnecessary and could be easily removed or placed in Supplementary files.
We understand the reviewer’s concern regarding the manuscript length. However, we do not believe any sections in the manuscript are redundant or unnecessary. This view is also supported by the other three referees, who, after we addressed their detailed suggestions, recommended the manuscript for publication without reservations. We have carefully considered the content and structure of the manuscript to ensure it provides relevant and necessary information, and we believe the current length is appropriate to convey the complexity of the subject matter.
Additionally, we disagree with the suggestion that adding extra references would not significantly lengthen the manuscript. It would indeed extend the length, and since some of the suggested references are only loosely related to the content of our article, the increase in length would be considerable. Reviewer 4 recommends cutting the manuscript by at least half, which would result in a substantially different article—one that, in our view, would likely lose important context and depth. At the very least, this would undermine the efforts of the other three referees, whose valuable input significantly shaped and improved the current version of the article. If we were to essentially discard the current manuscript and write a completely new one, their contributions would be disregarded, and the resulting article would be very different from the one they unanimously recommended for publication.
- The introduction (almost 3 pages!) is just a serial, hectic and disjoint list of studies selected without an apparent criterium and more suitable for a discussion of scoping review article than for a research paper.
We acknowledge the reviewer’s concern regarding the introduction but believe that its content and structure are appropriate for the study. The introduction provides readers with the necessary context to understand the background and significance of our research and situates its findings within the broader body of knowledge. At 1,277 words, its length was consistent with the standard for research articles of this type and reflects the complexity of the subject matter. The referee seems to have overlooked that they are not the sole evaluator of the paper. The introduction (like all other sections of the manuscript) has the length and content it currently has because of the feedback provided by the other three referees, who explicitly requested certain changes and additions and approved the manuscript in its current form. Making significant changes now to accommodate this reviewer’s suggestion—by removing parts of the introduction or substantially altering its structure—would effectively create a version of the article that the other referees have not reviewed or approved. Publishing such a fundamentally different manuscript without their oversight would, in our view, be ethically problematic.
- This is a study on just 584 individuals, of whom just 4 (=1.4%) hospitalized, with results not enough controlled for residual confounders (just sex, age and survey year). The authors stressed on the cross-sectional design as the main limitation, but this is not the main weakness of this study. The main limitation is the poor control for residual confounders (just age. Sex and survey year). Controlling for higher number of potential confounders would definitely help ruling out some spurious associations.
We regret to say that we must once again disagree with this comment. As outlined in the article, this study consists of three separate studies (with the overall study design being a serial cross-sectional study) involving 724 participants. The results presented in the first part of the Results section (the incidence of diagnosed and undiagnosed COVID-19 among female and male students) are based on the entire sample, not just the subset mentioned by the reviewer.
We also do not agree with the assertion that insufficient control for confounders is an important limitation of our study. In addition to controlling for age, sex, and survey year in the statistical analyses, the design of the study inherently minimized the influence of many other potential confounders. All participants shared the same temporary residence (Prague), the same racial background (Caucasian Europeans), the same education level (undergraduate students), and likely had very similar interests as they were all studying biology and enrolled in an elective course in evolutionary biology. Given this high level of homogeneity, we believe the risk of confounding by unmeasured variables is significantly lower compared to many other similar studies. Moreover, the reviewer does not provide any specific examples of additional confounders that should have been considered.
While preparing this response, we recognized that international readers might not fully appreciate the homogeneity of the population in the post-socialist Czech Republic and the exceptionally uniform characteristics of the participants in our study. To address this, we have now added a mention of this uniformity and its implications for potential confounding variables in the limitations section of the paper:
“In this study, we controlled for only three variables: age, survey year, and sex. Naturally, both health and cognitive performance are influenced by numerous additional confounding factors that were not examined or included in our statistical models. It is important to emphasize that the population of the Czech Republic, and particularly the participants in this study, is exceptionally homogeneous. Nearly all participants resided in Prague, shared a Caucasian European racial background, and belonged to a similar age group. They also had comparable educational qualifications as undergraduate biology students enrolled in an elective course on evolutionary biology, likely driven by similar motivations and interests. This high degree of homogeneity significantly minimized variability that could introduce confounding effects. However, it simultaneously limits the generalizability of the findings.”
- The authors mentioned some Hills criteria for causality, but biological plausibility is mainly founded on hypotheses and speculations and reproducibility of results may not be feasible yet biased by limited generalizability.
We strongly disagree with this comment. The passage in the Discussion section that Referee 4 refers to as “hypotheses and speculation” specifically states: “Plausibility (Criterion 6): It is more likely that SARS-CoV-2 infection worsens health rather than improves it.” As Karl Popper argued, science does not deal in absolute facts but rather in hypotheses that have withstood varying degrees of rigorous attempts at falsification, making them more or less credible. In this case, the hypothesis that SARS-CoV-2 infection is more likely to worsen health than improve it has been extensively tested and is supported by a substantial body of evidence. Suggesting otherwise disregards this evidence and undermines one of the most well-established conclusions in current biomedical research.
- Methods The FDR set at 10% for Benjamini Hichberg is rather loose criterium. It should be set at 5%
Once again, we strongly disagree with this comment. The referee seems to conflate the concept of statistical significance (alpha, typically set at 5% to reflect the tolerable risk of a Type I error) with the false discovery rate (FDR), which is an entirely different concept. Unlike methods such as the Bonferroni correction that control the overall family-wise error rate (FWER), the Benjamini-Hochberg (BH) procedure controls the expected proportion of false positives among the results identified as significant.
To clarify, if 50 tests are performed and BH correction identifies 10 tests as significant with an FDR threshold of 0.1 (10%), we can reasonably conclude that approximately 9 of these associations are genuinely significant, while one is likely a false positive. Importantly, this does not mean that any particular association—such as the one with the lowest p-value—is false. Instead, it reflects the average proportion of false discoveries across all significant results.
Setting the FDR threshold at 0.05 would only be meaningful (but not very practical) if we identified a large number of significant associations—20 or more. In such cases, we could conclude that one or fewer of those 20 associations is likely false. Generally, it is recommended to set FDR values between 0.1 and 0.25, with the main guideline being the number of significant tests after BH correction. For example, if only 4 associations are significant after BH correction, setting the FDR to 0.25 is appropriate because the result can be interpreted as indicating that one of the 4 significant tests is likely not actually significant. On the other hand, it would not make sense to apply an FDR threshold of 0.05, as this would imply that a fraction, in this case 0.2, of four significant results is false, which lacks meaningful interpretation.
In my methodology lectures, I recommend an initial FDR threshold of 0.2 as an appropriate starting point, which can be adjusted if necessary. In power analysis, it is standard practice to aim for an 80% probability of detecting a true effect (leaving a 20% chance of a Type II error). Symmetrically, it should also be acceptable to allow a 20% chance of identifying a non-existent effect (false positive) when controlling for multiple tests. This ensures a pragmatic approach that balances Type I and Type II errors while maximizing the likelihood of detecting meaningful associations.
For a discussion of the theoretical background of the method, the relation between FDR and P-value, and the superiority of controlling FDR over other methods of eliminating multiple test artifacts, kindly refer to
McDonald JH. Handbook of Biological Statistics. 3rd edn. Baltimore, MD: Sparky House Publishing, 2014
Nakagawa S. A farewell to Bonferroni: the problems of low statistical power and publication bias. Behav Ecol 2004;15:1044–5.
- I do not believe COVID-19 is a problem in children and young individuals, unless developing severe disease. The authors acknowledge this point, but than they affirm that long-COVID-19 is independent from severity of disease. For instance, just 183 children aged 1-17 died for COVID.19 in the entire USA (a country of almost 350 million inhabitants) during 2020-2022, 70% (2 thirds) having some pre-existing medical conditions. (PMID: 39484882). This is a clear indication on the impact of COVID-19 in children.
The conclusions of scientific articles should be based on evidence, not on belief or personal wishes. As our data have shown and as we explained in the article, the health risks of acute COVID-19 do not correlate with the health risks of long COVID. Therefore, children's resistance to acute COVID-19 does not necessarily imply that young adults (or children) are also resistant to the long-term effects of COVID. Our results (as well as the results from other studies we cite in the article) suggest that the long-term impacts of COVID-19 might be more severe in younger individuals compared to older ones. Suppressing such conclusions in the article would mean subordinating science to ideology.
- Line 571-572: the lower use of medications in males is consistent with the literature, reporting higher prevalence of long-COVID-19 among females, especially with pre-existing depression, suggesting also a possible psychological yet hypochondriac underlying component [e.g. PMID: 36676046).
This point is already addressed in the article, where we cite the exact study suggested by the referee (PMID: 36676046). It seems the referee may have overlooked this, and the comment might have been included here inadvertently.
- 674-679 and Line 735-736: this is the more appropriate conclusions in the entire study, which is in contrast with comments on lines 870-871 (in conclusions)
We partially agree with this comment. Lines 735-736, “However, even in our sample of young individuals, the severity of the illness had a strong impact,” is one of the most important conclusions of our study. This is why it is mentioned in the second sentence of the Conclusions section:
“Our study suggests that contracting COVID-19, especially when accompanied by severe symptoms, may negatively affect health and cognitive performance. It also indicates that even among young individuals, who mostly experienced relatively mild cases of COVID-19, some symptoms (such as frequent fatigue, impaired memory, and certain health issues) could persist for up to four years.”
On the other hand, we do not believe that lines 674-679 belong in the conclusions of the study:
“When discussing the varying effects of different virus variants, it is important to reiterate that our study was conducted on a young student population, where the illness generally presented mildly, as evidenced by the fact that only four students (0.7%) required hospitalization due to COVID-19. It’s possible, and indeed likely, that the impact of different virus variants on individuals in older age groups, particularly those who experienced severe or very severe illness, would differ significantly from our observations.”
This paragraph merely reminds the reader that our study, titled “Persistent Health and Cognitive Impairments Up to Four Years Post-COVID-19 in YOUNG STUDENTS: The Impact of Virus Variants and Vaccination Timing,” focuses on the long-term effects of COVID-19 on the health and cognition of young people, who, with few exceptions, experienced only mild forms of the disease. It emphasizes that the conclusions cannot be generalized to older individuals, who often experience more severe illness. This paragraph is certainly an important part of the discussion but is not one of the study’s conclusions.
At the same time, we believe that lines 870-871, “These differences do not appear to be strongly related to variations in the clinical course of the original acute illness or the time elapsed since COVID-19,” belong in the conclusions. This represents an unexpected and, from a clinical medicine perspective, probably very important outcome of our study. We may have been the first to demonstrate that there is likely no direct correlation between the severity of acute COVID-19 illness caused by different SARS-CoV-2 variants and the severity of the long-term effects associated with those same variants. For example, variants that cause more severe acute illness (e.g., Delta) may result in fewer long-term effects than variants that cause milder acute illness (e.g., the original Wuhan strain). This finding is novel and has important implications for understanding the long-term consequences of SARS-CoV-2 infection.
- Line 713-714: this is another key point, I,e, limited generalizability
Yes, this is indeed a key point. The small number of males who were infected for more than 3 years (15) somewhat reduces the significance of the result suggesting that, by the fourth year after contracting COVID-19, the condition of at least the male participants starts to improve. Due to the small sample size in this group, we did not include this particular result in the conclusions or the abstract; we only presented it in the results section and discussed it in the Discussion chapter.
- Line 717-718: vaccination has a 95% efficacy against development of severe disease, which is the key risk factor for long-COVID-19 ! this point again questions the generalizability of results, which may be biased by unmeasured factors. In fact, vaccination before COVID-19 was associated with higher cognitive performance compared to vaccination after COVID-19 in this study.
We fully agree that vaccines provide critical protection against severe COVID-19. However, it is also well-documented that vaccinated individuals often show worse health outcomes in cross-sectional studies compared to unvaccinated individuals, a pattern that has frequently been misinterpreted by anti-vaccine advocates. This positive association between vaccination and illness is largely due to individuals with poorer health or higher pre-existing illness rates being more concerned about infection and, consequently, more likely to seek vaccination. In our previous study of 4,445 individuals, we confirmed that vaccinated individuals who contracted COVID-19 reported higher rates of pre-existing illness prior to their infection compared to unvaccinated individuals who also contracted COVID-19. However, at the time of the study, their immediate health was better than that of the unvaccinated group who had experienced the disease.
In the current study, we similarly demonstrated—by comparing the health and cognition of individuals vaccinated before and after COVID-19—that the observed association between vaccination and worsened health is driven by pre-existing conditions rather than vaccination itself. This point is discussed in detail in the manuscript.
That said, we are uncertain how the established efficacy of vaccination in preventing severe COVID-19 could directly raise concerns about the generalizability of our results. Our study's design and analysis take these factors into account, and the conclusions remain robust.
Line 750-751: I believe the most important factor to explain such results may be poor control for residual confounders.
It is possible, though unlikely, that the observed results can be attributed to poor control for residual confounders. Similar findings have been reported in several previous studies conducted on different populations, which supports the robustness of our results. Additionally, we emphasize that the homogeneity of the studied population, combined with statistical control for the three key confounding variables (age, sex, and survey year), minimizes the risk of significant bias from unknown confounders. This level of control and population uniformity likely reduces the likelihood of confounding to a degree greater than that in the majority of comparable studies, a consideration that the referee may not have fully accounted for.